# Vascular smooth muscle RhoA counteracts abdominal aortic aneurysm formation by modulating MAP4K4 activity

Md Rasel Molla [1], Akio Shimizu[1], Masahiro Komeno[1], Nor Idayu A. Rahman[1], Joanne Ern Chi Soh[1], Le Kim Chi Nguyen[1], Mahbubur Rahman Khan[1], Wondwossen Wale Tesega [1], Si Chen[1,2], Xiaoling Pang[1,2], Miki Tanaka-Okamoto[3], Noriyuki Takashima[4], Akira Sato[1], Tomoaki Suzuki[4] & Hisakazu Ogita [1✉]

Whether a small GTPase RhoA plays a role in the pathology of abdominal aortic aneurysm (AAA) has not been determined. We show here that RhoA expression is reduced in human AAA lesions, compared with normal areas. Furthermore, incidence of AAA formation is increased in vascular smooth muscle cell (VSMC)-specific RhoA conditional knockout (cKO) mice. The contractility of the aortic rings and VSMCs from RhoA cKO mice is reduced, and expression of genes related to the VSMC contractility is attenuated by loss of RhoA. RhoA depletion activates the mitogen-activated protein (MAP) kinase signaling, including MAP4K4, in the aorta and VSMCs. Inhibition of MAP4K4 activity by DMX-5804 decreases AAA formation. Set, a binding protein to active RhoA, functions as an activator of MAP4K4 by sequestering PP2A, an inhibitor of MAP4K4, in the absence of RhoA. In conclusion, RhoA counteracts AAA formation through inhibition of MAP4K4 in cooperation with Set.

[1] Division of Molecular Medical Biochemistry, Department of Biochemistry and Molecular Biology, Shiga University of Medical Science, Otsu, Japan.
[2] Department of Emergency, The Fourth Affiliated Hospital of China Medical University, Shenyang, China. [3] Department of Molecular Biology, Osaka International Cancer Institute, Osaka, Japan. [4] Division of Cardiovascular Surgery and Thoracic Surgery, Department of Surgery, Shiga University of Medical Science, Otsu, Japan. ✉email: hogita@belle.shiga-med.ac.jp

Abdominal aortic aneurysm (AAA) causes aortic rupture and sudden death, and thus, is a life-threatening vascular disease[1]. AAA is formed by several factors, such as alteration of vascular smooth muscle cell (VSMC) force generation, the degradation and fragmentation of elastic fibers, loss of medial VSMCs, and inflammation[2,3]. Impaired contractility of VSMCs may be a major promoting factor for AAA. VSMCs of AAA patients have recently been shown to have impaired maximum contraction, compared with non-pathologic control VSMCs[4]. To respond to mechanical stress, VSMCs express high levels of contractile proteins, such as α-smooth muscle actin (α-SMA or ACTA2) and smooth muscle myosin heavy chain 11 (MYH11), and these cells are essential for the stabilization of aortic wall. Furthermore, the contractile proteins and their related molecules contribute to force distribution in the aorta by regulating the mechanical properties through the link with the extracellular matrix (ECM)[5]. In contrast, reduced VSMC force generation enhances matrix metalloproteinases (MMPs) activity and inflammatory cytokines/chemokines secretion[6–8]. To prevent AAA formation, keeping VSMCs healthy is considered important. However, how the VSMC contractile ability is regulated is not completely clear.

RhoA is a GTPase that is abundantly expressed in VSMCs[9,10]. The activity of RhoA is regulated by its cycling between the active GTP-bound form and the inactive GDP-bound form[11]. To activate RhoA, guanine nucleotide exchange factors induce the release of GDP from RhoA, resulting in the capture of GTP from the cytosol. Activated RhoA then transduces signals to downstream effectors[12]. For RhoA inactivation, GTPase-activating proteins (GAPs) robustly stimulate the intrinsic hydrolysis activity of RhoA to hydrolyze GTP to GDP. RhoA has multiple functions in regulating VSMCs and contributes to the morphological support and maintenance of the aortic structure[13]. Rho-associated coiled-coil containing kinase (ROCK), a molecule downstream of RhoA, mediates most of the functions of RhoA, including actin dynamics, cell contraction, and cell motility[14]. Dysfunctions of RhoA/ROCK have been linked to the pathogenesis of arteriosclerosis, ischemic injury, pulmonary hypertension, and heart failure, leading to the development of cardiovascular diseases[15]. The role of RhoA in the vasculature through ROCK has vigorously been investigated. However, how RhoA itself in VSMCs regulates the homeostasis of the aorta remains unclear.

Based on the background described above, we examined the RhoA expression levels in the normal and AAA areas of patients with AAA, and found that the level was significantly reduced in the medial layer of the AAA area. To further explore the function of RhoA in aortic VSMCs, VSMC-specific RhoA conditional knockout (cKO) mice were generated by the Cre/loxP system. Pharmacological stimulation highly induced AAA formation in RhoA cKO mice, compared with control mice. Loss of RhoA in VSMCs significantly decreased the contractility of the aortic rings and VSMCs and the expression of genes related to the contractility, and it increased the inflammatory response and endothelial injury, leading to the proteolytic and inflammatory VSMC phenotypes. We demonstrated that the inhibition of mitogen-activated protein kinase kinase kinase kinase 4 (MAP4K4) by RhoA in VSMCs was important for the prevention of AAA formation. Our proteomics analysis further helped to clarify the underlying molecular mechanism by which RhoA negatively regulated MAP4K4.

## Results

**Reduced expression of RhoA in patients with AAA.** Before examining RhoA expression in the human AAA, we confirmed the disrupted medial layer and loss of intact medial elastic fibers in the aortic wall of AAA patients by hematoxylin and eosin (H-E)

and elastin-specific Verhoeff Van Gieson staining (Fig. 1a, b). Immunohistochemistry and immunostaining of the aorta revealed that RhoA expression was markedly reduced in the medial layer of the AAA area compared with the normal area (Fig. 1c–f). In agreement with these results, qPCR and western blotting of aortic wall samples showed significantly decreased expression of RhoA mRNA and protein in the AAA area (Fig. 1g–i). These observations indicate that RhoA expression is reduced in human AAA and suggests that reduced RhoA expression may be involved in the development of AAA. In addition, the expression levels of VSMC and endothelial cell marker proteins α-SMA and CD31, respectively, were reduced in the AAA area due to the disruption of both medial and endothelial layers of the aortic wall, while the level of fibroblast marker protein vimentin was similar between the normal and AAA areas (Fig. 1h, i).

**Basal characteristics of RhoA cKO mice.** To determine whether impaired RhoA expression in the medial layer of the aorta contributes to the development of AAA, we generated VSMC-specific RhoA cKO mice using the Cre/loxP system. RhoA cKO mice displayed normal growth and were born in Mendelian ratios; the mice did not exhibit any gross phenotypic differences, including in fertility, compared with littermate controls. We confirmed absence of RhoA expression in the medial layer of the RhoA cKO mouse aorta by immunofluorescence staining (Supplementary Fig. 1a). Hemodynamic study showed a similar heart rate and systolic blood pressure (BP) between control and RhoA cKO mice at least by a spot check using the tail-cuff method (Supplementary Fig. 1b). The external appearance of the aorta was clear in both control and RhoA cKO mice, and AAA was not formed (Supplementary Fig. 1c). Ultrasonographic observation did not detect significant dilatation or difference in abdominal aortic diameter between control and RhoA cKO mice (Supplementary Fig. 1d, e). In addition, H-E and Verhoeff Van Gieson staining showed intact aortas in both control and RhoA cKO mice (Supplementary Fig. 1f, g). Tropoelastin, encoded by the *Eln* gene, and fibrillin-1, encoded by the *Fbn1* gene, are the major components of the medial layer of the aorta and play a role in the regulation of aortic elasticity[16]. The levels of *Eln* and *Fbn1* mRNAs were similar in the aorta between both types of mice (Supplementary Fig. 1h). We further examined collagen production and collagen cross-linking in the mouse aorta samples, which may affect the strength of the aortic wall, by qPCR and hydroxyproline assay. The mRNA level of collagen type I α1, the major component of type I collagen, was similar between control and RhoA cKO mice aortas. The hydroxyproline assay also showed that total collagen and cross-linked collagen were equivalently contained in the aortas of both types of mice (Supplementary Fig. 1i, j). These results suggest that in normal conditions, loss of RhoA in VSMCs does not affect the morphology and function of the aorta at least during the six months after birth.

It has been reported that the activity of Cre recombinase driven under the SM22α promoter was observed in the heart as well as in VSMCs in mice[17], and that the target gene inactivation was shown in the heart of such cKO mice[18]. Based on these previous publications, we checked the expression of RhoA in the heart of embryos and adult mice. The RhoA expression was actually reduced, but not completely lost, in both developing and adult hearts of RhoA cKO mice, compared with those of control mice (Supplementary Fig. 2a–d). Despite the reduced expression of RhoA in the heart, the cardiac contractility and dimensions in RhoA cKO adult mice were maintained (Supplementary Fig. 2e), suggesting that a degree of reduction of cardiac RhoA expression in RhoA cKO mice does not affect the heart function.

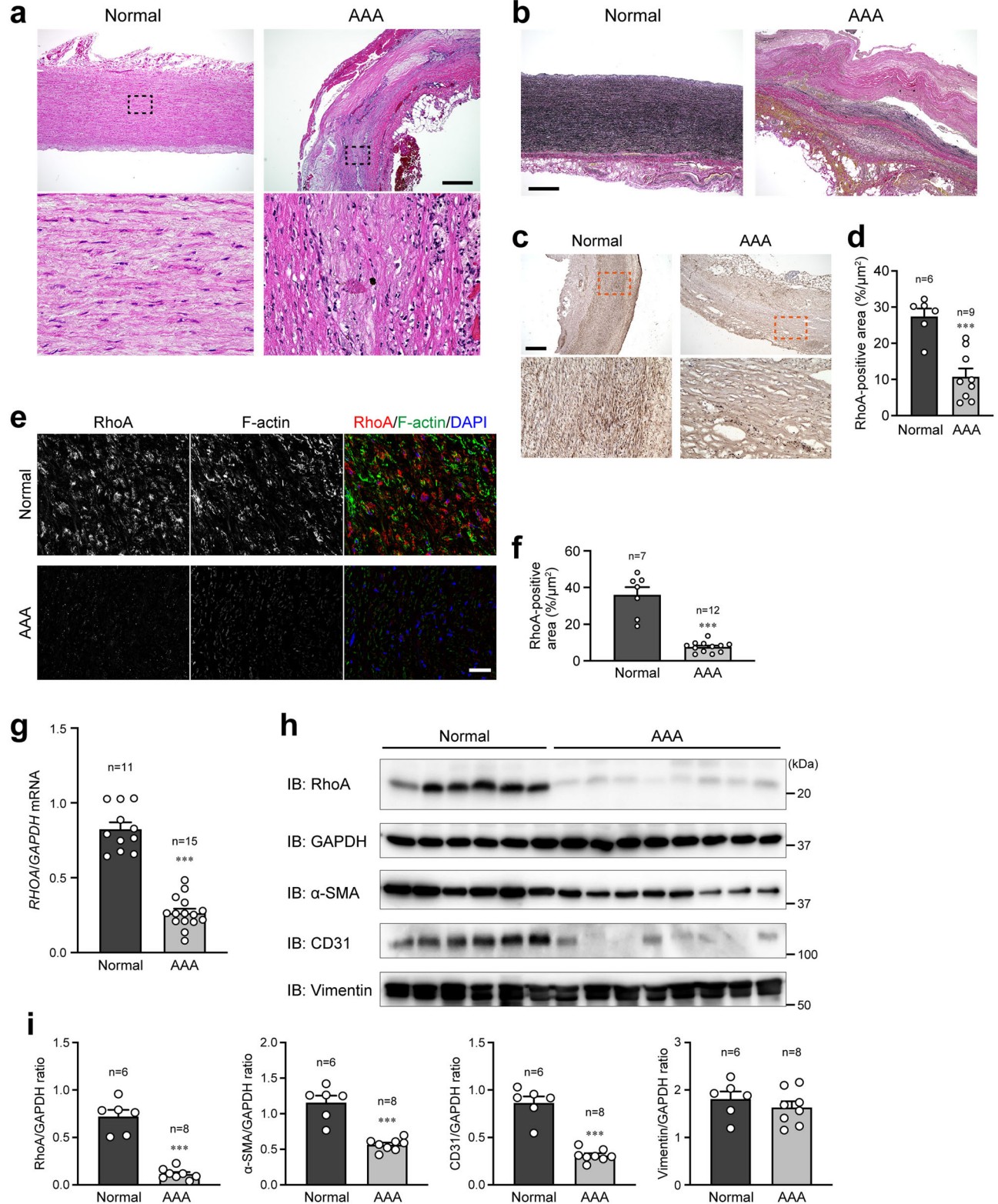

**Induction of AAA by pharmacological stimulation**. Based on the results of the RhoA cKO mice described above, we next examined the biological significance of RhoA in VSMCs under pharmacological stimulation. Angiotensin II (AngII) and β-aminopropionitrile (BAPN) were administered to control and RhoA cKO mice using osmotic mini-pumps for 4 weeks to induce AAA. No differences in body weight, heart rate, and systolic BP measured by the tail-cuff method were observed between the groups during AngII and BAPN administration (Fig. 2a). However, the incidence of AAA formation was significantly higher in RhoA cKO mice than in control mice (Table 1 and Fig. 2b). We further analyzed control mice in which AAA was formed, and found that the RhoA expression was remarkably reduced in the AAA area compared with the

**Fig. 1 Reduced RhoA expression in the human abdominal aortic aneurysm. a** Hematoxylin and eosin (H-E) staining of normal and abdominal aortic aneurysm (AAA) areas in the human aorta. Dotted rectangles are enlarged and shown in the bottom. **b** Verhoeff Van Gieson staining of normal and AAA areas. **c** Immunohistochemistry of RhoA in normal and AAA area of the aorta. Dotted rectangles are enlarged and shown in the bottom. **d** Summary graph of RhoA expression examined in (**c**). $n = 6$ in Normal and $n = 9$ in AAA. **e** Immunostaining of RhoA in normal areas and AAA lesions. F-actin and nuclei were counterstained with phalloidin and DAPI, respectively. **f** Summary graph of RhoA expression examined in (**e**). $n = 7$ in Normal and $n = 12$ in AAA. **g** qPCR analysis for *RHOA* mRNA obtained from normal and AAA areas. *GAPDH* was used as a control. $n = 11$ in Normal and $n = 15$ in AAA. **h** Western blotting for RhoA and other marker proteins in normal and AAA areas. GAPDH served as the loading control. **i** Summary graphs of relative expression of RhoA and other marker proteins examined in (**h**). $n = 6$ in Normal and $n = 8$ in AAA. Scale bars: 500 μm (**a–c**) and 30 μm (**e**). In (**d**, **f**, **g**, **i**), the data between the two groups were analyzed by $t$-test. ***$p < 0.001$ vs. Normal.

normal area (Supplementary Fig. 3a–c), suggesting the importance of RhoA for prevention of AAA. Although the length of the aorta was not different between control and RhoA cKO mice (Fig. 2c), the diameter of the abdominal aorta measured by ultrasonography was significantly larger in RhoA cKO mice compared with control mice (Fig. 2d, e). Histological analysis by H-E and Verhoeff Van Gieson staining showed that the disruption and degradation of medial elastic lamina in the aorta were more severe in RhoA cKO mice than control mice (Fig. 2f–h). In addition, *Eln* and *Fbn1* mRNA levels were significantly reduced after Ang II and BAPN infusion in RhoA cKO mice compared with control mice (Fig. 2i). Previous studies have suggested that an abnormal increase in proteoglycan aggregation in the aorta can cause disruption of cell-matrix interactions, compromised aortic mechanosensing and structural integrity, and may be a hallmark of medial degeneration in the aorta[19–21]. Therefore, we performed immunostaining to detect aggrecan expression in the aorta and found that the expression was significantly up-regulated and accumulated at the site of aortic rupture in RhoA cKO mice (Fig. 2j). Together these results suggest that RhoA in VSMCs has a protective role against the stimulation to form AAA.

**Effect of RhoA in VSMCs on vascular contractility and VSMC phenotype.** We next examined the role of RhoA in VSMCs in the contractility of the aorta and VSMCs. After infusion of saline or both AngII and BAPN for 4 weeks, the isometric tension of the isolated mouse aorta was measured by a wire myograph. The aortas from control mice resisted the highest tension generated by the multi-wire myograph to similar levels after the infusion of saline and AngII+BAPN, while the aortas from RhoA cKO mice were fragile and preserved the lower aortic tension regardless of treatment with saline or AngII+BAPN (Fig. 3a). We also performed in vitro collagen gel contractility assay using isolated VSMCs from the control and RhoA cKO mouse aorta. We found that independent of AngII treatment, the reduction of the diameter of the collagen gel containing RhoA cKO VSMCs was significantly impaired, compared with that of the gel containing control VSMCs (Fig. 3b, c). These results indicate that RhoA cKO VSMCs have less contractility than control VSMCs.

Then, we checked the control and RhoA cKO aortas and VSMCs by examining markers for VSMC contractile structure, such as α-SMA[6]. Immunofluorescence and western blotting analyses showed that α-SMA expression was decreased in the RhoA cKO aorta and VSMCs, compared with control ones (Fig. 3d–g). In addition, reduced and disorganized expression of F-actin was observed in response to loss of RhoA in the aorta and VSMCs (Fig. 3d, e). The reduced expression of α-SMA and impairment of F-actin were also found in the AAA area of the control aorta (Supplementary Fig. 3b). qPCR revealed significantly reduced expression of other contractile marker genes, *Mylk* and *Myh11* as well as *Acta2*, in RhoA cKO VSMCs compared with control VSMCs (Fig. 3h). We further performed the rescue experiments to confirm that the reduced expressions of α-SMA and

other VSMC contractile marker genes were specifically dependent on loss of RhoA in VSMCs. When GFP-RhoA-wild-type (WT) was re-expressed in RhoA cKO VSMCs at the level like control VSMCs, α-SMA protein expression was completely recovered, and normal F-actin organization was observed in the cells (Supplementary Fig. 4a–c). Similarly, reduced expression of contractile marker genes in RhoA cKO VSMCs was restored by re-expression of GFP-RhoA-WT (Supplementary Fig. 4d).

**Decreased endothelial barrier and increased inflammatory cell infiltration in the aorta of RhoA cKO mice.** Because dysfunction and disorganization of the medial layer of the aorta usually affect endothelial barrier function and vice versa[22,23], we examined the morphology of the endothelial layer and found that the layer was disrupted mainly at the site of AAA in RhoA cKO mice after AngII+BAPN treatment (Fig. 4a). This impaired endothelial barrier function in the RhoA cKO aorta was further confirmed by dye accumulation in the aortic wall. When Evans Blue dye was intravenously injected in the mice, the dye accumulation was significantly increased in AngII+BAPN-treated RhoA cKO mice (Fig. 4b, c). The disruption may allow inflammatory cells to infiltrate into the aortic wall. Immunofluorescence staining showed that in the saline infusion, inflammatory cell infiltration as determined by CD68 and F4/80 markers was minimal in both control and RhoA cKO aortas, but in response to treatment with AngII+BAPN, the infiltration was markedly increased in RhoA cKO aorta (Fig. 4d, e). Inflammatory cytokines facilitate endothelial layer disruption and inflammatory cell infiltration[24]. We next investigated the expression of inflammatory cytokines and found that it was increased in the RhoA cKO aorta after a 4-week infusion of reagents and in isolated RhoA cKO VSMCs after AngII stimulation (Fig. 4f, g). In addition, the increase in the expression of inflammatory cytokines in RhoA cKO VSMCs was reversed by transfection of GFP-RhoA-WT in the cells (Supplementary Fig. 4e). These results suggest that loss of RhoA in aortic VSMCs enhances vascular inflammation via the up-regulation of cytokine production.

**Expression of molecules that regulate extracellular matrix degradation of the aorta in loss of RhoA.** To further investigate the potential mechanisms for the increased elastin destruction in the RhoA cKO aorta, we examined the expressions of MMP2 and MMP9, which are key proteinases in extracellular matrix (ECM) protein destruction. *Mmp2* mRNA level in AngII-stimulated cultured VSMCs was significantly higher in the RhoA cKO than the control (Supplementary Fig. 5a). Immunostaining showed that the MMP2 expression in the aorta was increased in RhoA cKO mice, compared with the control mice, after treatment with AngII and BAPN (Supplementary Fig. 5b, c). Similar results were obtained for MMP9 expression (Supplementary Fig. 5a, d, e). In contrast, the expressions of tissue inhibitor of metalloproteinase 1 (TIMP1) and TIMP2, which are endogenous inhibitors for MMPs, were markedly reduced in cultured VSMCs and aorta of

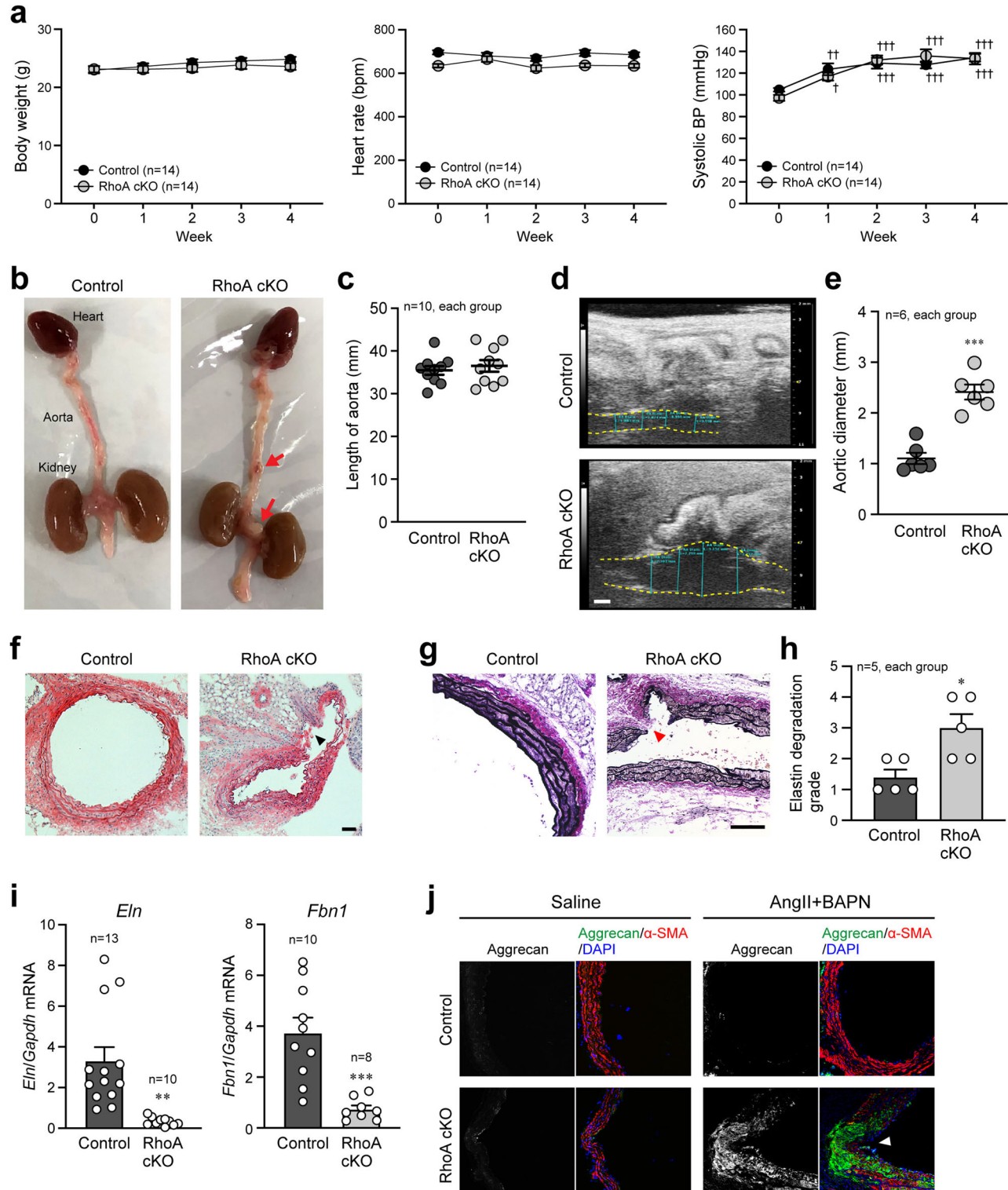

**Fig. 2 Induction of AAA in RhoA cKO mice treated with AngII and BAPN. a** Changes in body weight and hemodynamics before (Week 0) and after treatment with AngII and BAPN. $n = 14$, each group. **b** External appearance of the aorta with the heart and kidneys extracted after the 4-week treatment. Arrows indicate formed AAA. **c** Length of aorta from the aortic valve to the common iliac artery bifurcation. **d** Ultrasonographic observation of the abdominal aorta, which is outlined by yellow dotted lines, after the treatment. **e** Summary graph of the abdominal aortic diameter. $n = 6$, each group. **f** H-E staining of the aorta. **g** Verhoeff Van Gieson staining of the aorta. **h** Summary graph of the elastin degradation grade analyzed in (**g**). $n = 5$, each group. **i** qPCR analysis of genes related to aortic elasticity. $n = 13$ in Control and $n = 10$ in RhoA cKO for *Eln*, and $n = 10$ in Control and $n = 8$ in RhoA cKO for *Fbn1*. **j** Co-immunostaining of aggrecan and α-SMA in the mouse aorta after the treatment. Nuclei were counterstained with DAPI. Arrowheads: position of AAA rupture (**f**, **g**, **j**). Scale bars: 1 mm (**d**) and 100 μm (**f**, **g**, **j**). In (**a**), two-way or one-way ANOVA was applied to compare the data between groups and weeks, respectively, and in (**c**, **e**, **h**, **i**), the data between the two groups were analyzed by *t*-test. $^{\dagger}p < 0.05$, $^{\dagger\dagger}p < 0.01$, and $^{\dagger\dagger\dagger}p < 0.001$ vs. Week 0; $^{*}p < 0.05$, $^{**}p < 0.01$, and $^{***}p < 0.001$ vs. Control.

**Table 1 Number of mice with or without AAA after treatment with AngII and BAPN.**

|            | Control | RhoA cKO  |
|------------|---------|-----------|
| AAA ( + )  | 8 (4)   | 21*** (9) |
| AAA ( − )  | 18      | 5         |
| Total      | 26      | 26        |

***p < 0.001 vs. Control. Fisher's exact test was applied to examine the statistical significance between the groups.
Numbers in () indicate the deaths by rupture of AAA.

**Table 2 Number of mice with or without AAA after treatment with vehicle or DMX-5804 in addition to AngII and BAPN infusion.**

|           | RhoA cKO |          | Control  |
|-----------|----------|----------|----------|
|           | Vehicle  | DMX-5804 | DMX-5804 |
| AAA ( + ) | 8 (2)    | 1* (0)   | 2 (0)    |
| AAA ( − ) | 3        | 6        | 6        |
| Total     | 11       | 7        | 8        |

*p < 0.05 vs. Vehicle. Fisher's exact test was applied to examine the statistical significance between the groups.
Numbers in () indicate the deaths by rupture of AAA.

RhoA cKO mice (Supplementary Fig. 5f–j), suggesting that both increased MMPs expression and their enhanced activity from reduced expression of TIMPs cooperatively contribute to the robust degradation of ECM for AAA formation in the aorta of RhoA cKO mice. Zymography using human aortic tissue samples also detected higher amounts of pro and mature MMP2 and MMP9 in the AAA lesion than the normal area (Supplementary Fig. 5k, l). These data indicate that RhoA inhibits MMPs expression and activation, resulting in fewer ECM abnormalities in the aorta of control mice after pharmacological stimulation.

**Regulation of the mitogen-activated protein kinase signaling by RhoA.** We next sought to investigate the signaling pathway that is responsible for the inflammation and MMPs expression in VSMCs of the aorta. Previous studies have shown that mitogen-activated protein (MAP) kinase signaling factors, such as extracellular signal-regulated kinase (ERK) and p38, regulate the above phenomena[25,26]. Thus, we examined the activity (phosphorylation level) of ERK1/2 and p38. The immuno-fluorescence signal of phosphorylated ERK1/2 and p38 was increased in the RhoA cKO aorta, compared with the control aorta, after the 4-week AngII+BAPN treatment (Fig. 5a–d). This increase was also observed in VSMCs stimulated with AngII (Fig. 5e, f). To further explore the molecule that regulates the activity of MAP kinases, we focused on MAP4K4[27,28], which is also related to vascular inflammation and cardiomyocyte contractility[29,30]. Pharmacological stimulation-induced activation (phosphorylation) of MAP4K4 was significantly higher in the aorta and isolated VSMCs of RhoA cKO mice than in those of control mice (Fig. 5g–j). In addition, AngII-induced enhancement of MAP kinase signaling pathway in RhoA cKO VSMCs was returned to the level in control VSMCs by re-expression of GFP-RhoA-WT (Supplementary Fig. 4f, g). The specificity of antibodies for phosphorylated ERK1/2, p38, and MAP4K4 immunostaining was validated by experiments using the RhoA cKO aortas after inhibitor administration. The signals for phosphorylated ERK1/2, p38, and MAP4K4, which were

clearly detected by AngII+BAPN treatment, disappeared in the presence of the inhibitor of each MAP kinase signaling molecule (Supplementary Fig. 6). In the human aorta, activation of MAP4K4 was enhanced in the AAA lesion, compared with the normal area (Fig. 5k–n), and even in control mice, the enhanced activation was also observed at the site of AAA (Supplementary Fig. 3d, e). These results suggest that RhoA in VSMCs may attenuate the MAP kinase signaling cascade to prevent AAA formation.

We further evaluated the significance of MAP4K4 activity for AAA formation in the absence of VSMC RhoA using the MAP4K4 inhibitor DMX-5804. AAA formation was significantly reduced by intravenous treatment with DMX-5804 in addition to AngII and BAPN in RhoA cKO mice, with no change in hemodynamics (Table 2, Fig. 6a and Supplementary Fig. 7a). Histological analyses showed that DMX-5804 treatment reversed the adverse morphology and degradation of the RhoA cKO aorta similar to control aorta treated with DMX-5804 (Fig. 6b–d). It was also confirmed that the DMX-5804 treatment did not affect the cardiac functions and the plasma volumes in RhoA cKO mice as well as control mice (Supplementary Fig. 7b, c). We further found that the aortic tension of RhoA cKO mice was recovered by MAP4K4 inhibition with DMX-5804 (Fig. 6e). Collagen gel contractility assay using isolated VSMCs also showed similar results (Fig. 6f), and morphological abnormalities with reduced α-SMA expression and disturbed F-actin organization in VSMCs treated with AngII were reversed by DMX-5804 (Fig. 6g).

Since VSMC contractility is critically dependent on the phosphorylation level of myosin light chain 2 (MLC2), we therefore examined the level in VSMCs by western blotting. After AngII stimulation, phosphorylated MLC2 was increased in the control VSMCs, but no increase was detected in RhoA cKO VSMCs; treatment of RhoA cKO VSMCs with DMX-5804 restored the phosphorylation level of MLC2 similar to that of control VSMCs (Fig. 6h, i). The phosphorylation of MLC2 is regulated by the balance of function of two enzymes: myosin light chain kinase (MYLK), which induces the phosphorylation of MLC2, and myosin light chain phosphatase (MLCP), which dephosphorylates MLC2 and is strongly inhibited by ROCK[31,32]. In RhoA cKO VSMCs, MYLK expression was reduced, which was in line with the mRNA results in Fig. 3h, and phosphorylated MYLK (the inactivated form of MYLK) was increased even with the reduced expression of MYLK (Fig. 6j, k). The enhanced phosphorylation of MYLK by loss of RhoA was blocked by treatment with DMX-5804. Moreover, immunoprecipitation assays showed that MAP4K4 interacted with MYLK (Fig. 6l). These results suggest MAP4K4 phosphorylates and inactivates MYLK, leading to inhibition of MLC2 phosphorylation. In addition, we observed that the reduced expression of *Myh11* as well as *Acta2* in RhoA cKO VSMCs was recovered by DMX-5804 treatment (Fig. 6m). Together, these data indicate the involvement of MAP4K4 in the regulation of the VSMC contractility.

The inhibitory effect of DMX-5804 on inflammation and MAP kinase signaling was examined in the mouse aortic samples and VSMCs. Additional treatment with DMX-5804 significantly decreased the expression levels of F4/80 and MMP2 as well as the phosphorylation of MAP4K4 in the RhoA cKO mouse aorta (Supplementary Fig. 8a, b). Western blot analysis showed that inhibition of MAP4K4 with DMX-5804 in VSMCs diminished the AngII-induced activation of p38 and ERK1/2 signaling that is associated with vascular inflammation and ECM degradation (Supplementary Fig. 8c, d). Indeed, increased expression of inflammatory cytokines in RhoA cKO VSMCs after AngII treatment was significantly suppressed by

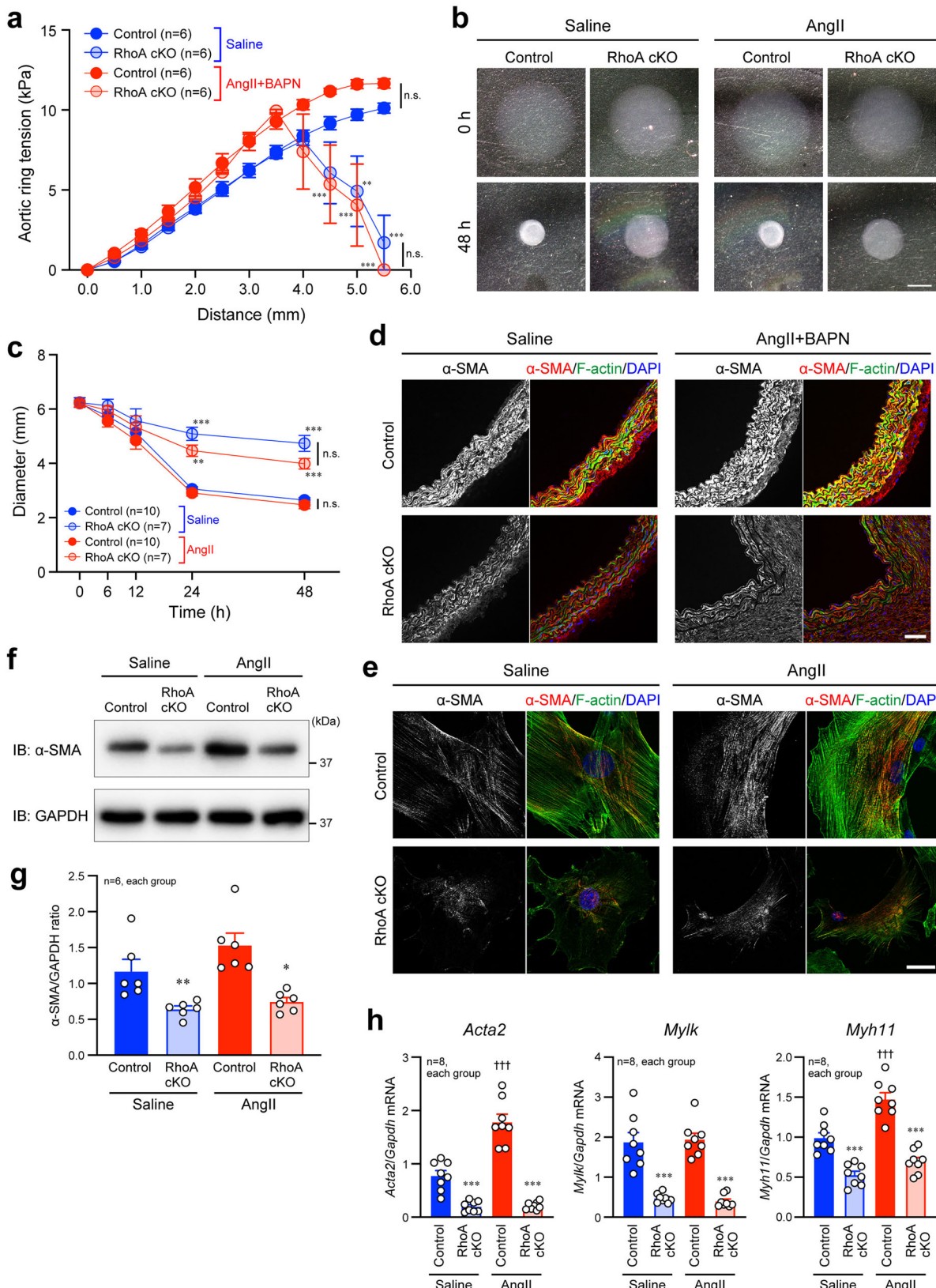

DMX-5804 (Supplementary Fig. 8e), and the AngII-mediated changes of MMPs and TIMPs expression in RhoA cKO VSMCs were completely canceled by DMX-5804 (Supplementary Fig. 8f). Together, these results indicate that inhibition of MAP4K4 in VSMCs lacking RhoA suppresses AAA formation, rescues the contractility of VSMCs, and attenuates vascular inflammation and ECM degradation through inactivation of the MAP kinase signaling.

**Involvement of Set and PP2A in RhoA-modulated MAP4K4 activity**. To explore the molecular mechanism by which RhoA

**Fig. 3 Contractility of aorta and aortic VSMCs and characteristics of VSMCs with or without expression of RhoA in VSMCs. a** Changes of aortic ring tension measured by a wire myograph. $n = 6$, each group. **b** In vitro collagen gel contractility assay using isolated mouse aortic VSMCs by treatment with saline or AngII. **c** Summary graph of VSMC-containing collagen gel diameter examined in (**b**). $n = 7-10$, each group. **d** Immunostaining of α-SMA in the mouse aorta after the 4-week treatment. **e** Immunofluorescence images of aortic VSMCs stained with the α-SMA antibody after treatment with saline or AngII for 24 h. F-actin and nuclei were visualized with phalloidin and DAPI, respectively, in (**d**, **e**). **f** Western blotting of α-SMA in aortic VSMCs after treatment with saline or AngII for 24 h. GAPDH was blotted for the loading control. **g** Summary graph of the α-SMA/GAPDH ratio examined in (**f**). $n = 6$, each group. **h** qPCR analysis of genes related to VSMC contractile phenotype. Aortic VSMCs were treated with saline or AngII for 24 h. $n = 8$, each group. Scale bars: 2 mm (**b**), 20 μm (**d**), and 50 μm (**e**). Two-way (**a**, **c**) or one-way (**g**, **h**) ANOVA was applied to compare the data between groups. *$p < 0.05$, **$p < 0.01$, and ***$p < 0.001$ vs. Control in the same treatment; †††$p < 0.001$ vs. Control in the saline treatment.

modulates MAP4K4 activity in VSMCs, we performed immunoprecipitation and mass spectrometry (MS). RhoA cKO VSMCs transfected with GFP-RhoA-wild-type (WT) or GFP control vector were stimulated with AngII, and then, the cell lysates were immunoprecipitated with anti-GFP antibody, followed by the detection of immunoprecipitants in Coomassie Brilliant Blue (CBB) staining (Fig. 7a). Proteins that co-immunoprecipitated with GFP-RhoA-WT, but not GFP, were analyzed by liquid chromatography (LC)-MS/MS. The analysis identified 1453 peptides that belonged to 359 proteins (Supplementary Data 1). Recently, MAP4K4 activity has been reported to be regulated by the striatin-interacting phosphatase and kinase (STRIPAK) complex, which is downstream of RhoA[33,34]. Based on this report, we performed in silico analysis for the 359 proteins to identify the protein that can preferentially interact with a component of the STRIPAK complex. We found that Set (Accession No. Q9EQU5) had the highest score for binding to protein phosphatase 2A (PP2A), a STRIPAK component that negatively regulates MAP4K4 activity by reducing the MAP4K4 phosphorylation. Previous studies also showed that Set associates with PP2A and inhibits its phosphatase activity[35,36], and that it regulates the signaling of Rac1, another Rho family GTPase[37].

Next, to determine whether Set actually contributes to the regulation of MAP4K4 activity, we performed knockdown experiments in RhoA cKO VSMCs with AngII stimulation, and found that Set siRNA transfection significantly reduced the phosphorylation of MAP4K4 (Fig. 7b, c). Then, we hypothesized that in response to AngII stimulation in VSMCs, the interaction of Set with AngII-induced activated RhoA would be increased, which dissociates Set from PP2A, and that free PP2A could bind to MAP4K4 and inhibit the activation of MAP4K4. Consistent with this hypothesis, the interaction of Set with GTP-bound RhoA was increased in VSMCs after AngII stimulation (Fig. 7d). In control VSMCs stimulated with AngII, the association of PP2A to MAP4K4 was observed, and this association was impaired by loss of RhoA because of PP2A–Set binding (Fig. 7e left panels and Fig. 7f), which leads to MAP4K4 autophosphorylation[38]. Furthermore, when Set was knocked down in RhoA cKO VSMCs, the association between PP2A and MAP4K4 was restored (Fig. 7e right panels). In contrast, when control VSMCs were transfected with HA-tagged Set isoform 1 and isoform 2 (HA-Set1 and HA-Set2) and stimulated with AngII, the PP2A–Set binding was increased and PP2A–MAP4K4 association was decreased even in the presence of RhoA (Supplementary Fig. 9a), suggesting the balanced and functional molecular interaction among Set, PP2A and MAP4K4 downstream of RhoA.

Finally, we confirmed that the Set-mediated regulation of MAP4K4 activation was involved in the activation of MAP kinases and VSMC contractility downstream of MAP4K4. Under the AngII stimulation, phosphorylation of ERK1/2 and p38 as well as MAP4K4 was not increased in RhoA cKO VSMCs by additional knockdown of Set (Fig. 7g, h). Consistently, the contractility of VSMCs was preserved, the phosphorylation of MLC2 was restored, and the phosphorylation of MYLK was not increased in RhoA cKO

VSMCs with siSet transfection (Fig. 7i–k). In AngII-stimulated control VSMCs transfected with HA-Set1 and HA-Set2, phosphorylated MAP4K4, ERK1/2, and p38 were significantly increased, compared with VSMCs transfected with an empty plasmid (Supplementary Fig. 9b, c). In addition, overexpression of HA-Set1 and HA-Set2 suppressed the phosphorylation of MLC2, and facilitated the inactivation (phosphorylation) of MYLK (Supplementary Fig. 9d, e), which may result from the increased activation of MAP4K4 by the abundant Set-mediated dissociation of MAP4K4 from PP2A. Collectively, these results suggest that RhoA negatively regulates MAP4K4 activation through Set and PP2A, resulting in the maintenance of contractility and morphology of VSMCs and the aorta.

## Discussion

In this study, we identified the protective effects of RhoA in VSMCs on formation of AAA using human aortic tissues, a RhoA cKO mouse model, and in vitro approaches. Loss of RhoA is involved in both dysfunction of VSMC contractility and enhancement of vascular inflammation. These conclusions are based on the following findings: (1) RhoA expression was remarkably reduced in human AAA lesion, (2) AAA formation was increased in RhoA cKO mice treated with AngII and BAPN, (3) VSMC contractility-related gene expression was reduced by loss of RhoA, and the diminished VSMC contractility with inflammatory cytokine expression was induced, (4) inflammatory cell infiltration and endothelial dysfunction were enhanced in RhoA cKO mouse aorta, (5) RhoA depletion induced ECM degradation as determined by increased MMPs activity and decreased TIMPs activity, and (6) the MAP4K4-mediated MAP kinase cascade was activated in RhoA-null aorta and VSMCs.

Although several reports showed the involvement of the regulators of RhoA, such as GAP or GDP dissociation stimulator for RhoA, in thoracic aortic aneurysm of which characteristics are different from AAA[39,40], RhoA itself has not been certified as a predisposing molecule to prevent AAA, which is unveiled in this study. We also found that RhoA in VSMCs was important for expression of force generation genes, such as *Acta2* and *Myh11*, and suppression of cytokines/chemokines gene transcription, such as *Il1b* and *Ccl2*. Thus, depletion of RhoA in VSMCs promoted the expression of such cytokines/chemokines and recruits inflammatory cells in the media of the aorta, in addition to the reduced contractility of VSMCs. Accumulated inflammatory cells in the aortic wall also produce abundant MMPs and enhance their activation, leading to the vulnerability of the wall[41]. These phenomena may critically contribute to the induction of AAA.

MAP4K4 in endothelial cells promotes vascular inflammation[30]. MAP4K4 acts as an upstream regulator of the MAP kinase signaling pathway, and depletion of MAP4K4 suppresses the pathway and inflammation[27,42]. In this study, we identified the association between RhoA and MAP4K4 in VSMCs and found that this association is important for the prevention of AAA. In the presence

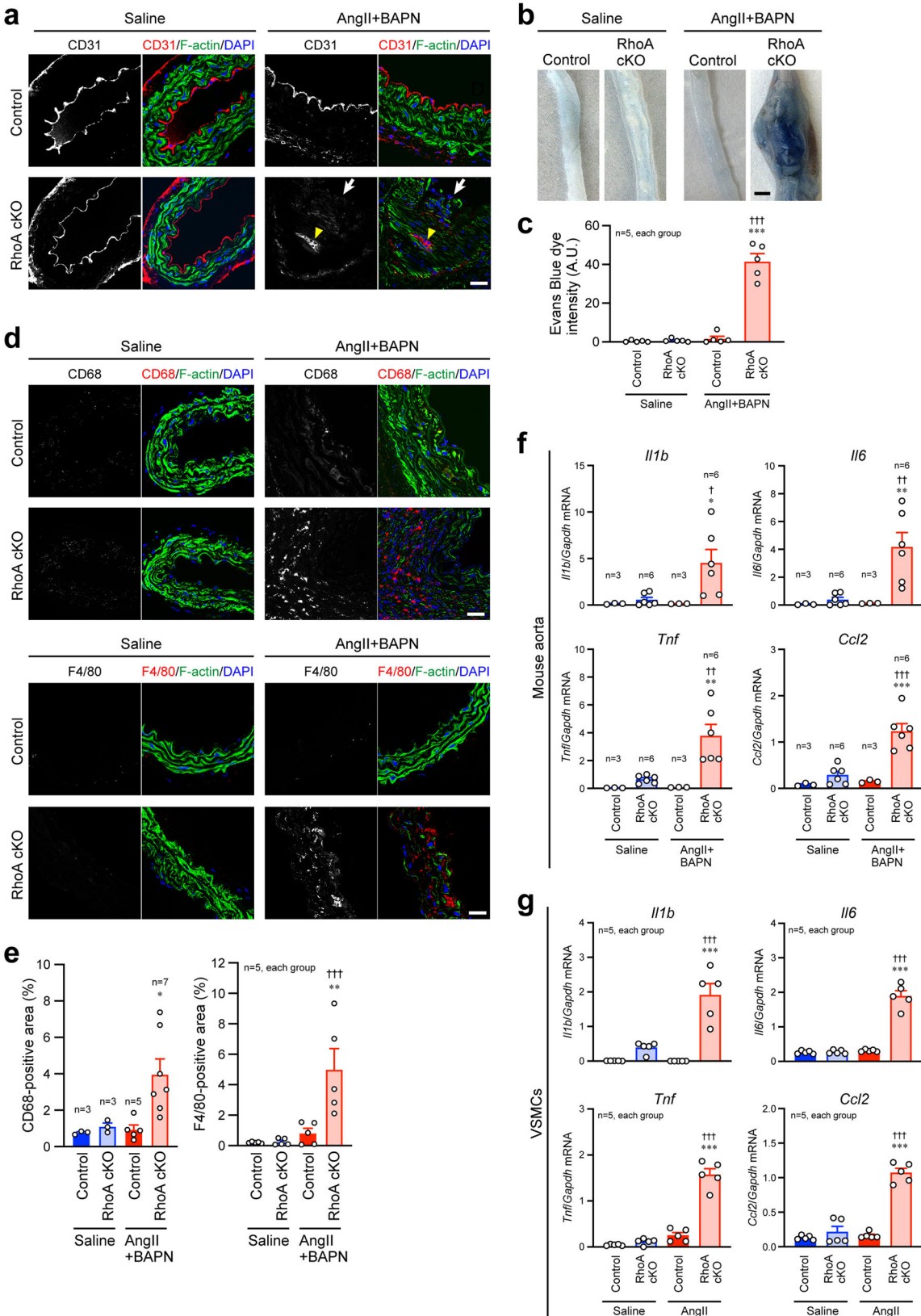

of RhoA, the activities of MAP4K4 and MAP kinases were inhibited, even after AngII stimulation, resulting in the reduction of inflammation. In contrast, in the absence of RhoA in VSMCs, MAP4K4 activity was up-regulated and the incidence of AAA formation was significantly increased. When MAP4K4 activity was attenuated by DMX-5804 in RhoA cKO mice and RhoA-depleted VSMCs, not only MAP4K4-mediated inflammatory response but also vascular contractility was clearly recovered by inducing MYLK and MLC2 activity as well as *Acta2* and *Myh11* expression. Moreover, the AAA formation induced by loss of RhoA was

**Fig. 4 Reduced endothelial barrier, increased inflammatory cell infiltration, and enhanced inflammatory cytokine expression in the RhoA cKO mouse aorta after treatment with AngII and BAPN. a** Immunofluorescence images of the control and RhoA cKO mouse aorta stained with the anti-CD31 antibody. The aorta was extracted after the 4-week treatment with saline or AngII+BAPN. F-actin and nuclei were visualized with phalloidin and DAPI, respectively. An arrow and an arrowhead indicate loss of endothelium and infiltration of inflammatory cells, respectively. **b** Vascular permeability assay to evaluate the endothelial barrier function in the aorta. At 10 min after the intravenous injection of Evans Blue dye, the aorta from mice treated with saline or AngII+BAPN was extracted, and the dye accumulation was measured. **c** Summary graph of mean intensity of Evans Blue dye accumulation. $n = 5$, each group. **d** Immunofluorescence images of the control and RhoA cKO mouse aorta stained with the CD68 or F4/80 antibody. The aorta was extracted as described in (**a**). **e** Summary graph of the percentage of CD68-positive area or F4/80-positive area in the media of the aorta. $n = 3-7$, each group. **f, g** Graphs of qPCR results for inflammatory cytokines in the mouse aorta (**f**) and aortic VSMCs (**g**). The aorta was extracted as described in (**a**), and VSMCs were treated with saline or AngII for 24 h. $n = 3-6$, each group. Scale bars: 30 µm (**a, d**) and 1 mm (**b**). In (**c, e, f, g**), one-way ANOVA was applied to compare the data between groups. $*p < 0.05$, $**p < 0.01$, and $***p < 0.001$ vs. Control in the same treatment. $^{†}p < 0.05$, $^{††}p < 0.01$, and $^{†††}p < 0.001$ vs. RhoA cKO in the saline treatment.

completely inhibited by DMX-5804 treatment. This shows that RhoA transduces the signal to MAP4K4 to inhibit its activity, and that the inhibition has a protective effect on AAA formation, suggesting that MAP4K4 may represent a therapeutic target against AAA.

A further finding of our study is the identification of a signaling molecule that links RhoA and STRIPAK, a molecular complex that modulates MAP4K4 activity. A previous study showed that active GTP-bound RhoA suppressed the activation (phosphorylation) of MAP4K4 through PP2A, a phosphatase in the STRI-PAK complex[34]. However, how RhoA regulates PP2A function has been unknown. We performed pull-down assay and LC-MS/MS using AngII-stimulated VSMCs to identify the molecule(s) that associates with both RhoA and PP2A. By the in silico analysis posterior to LC-MS/MS, Set was found to be a candidate for the association. We further confirmed the molecular association by immunoprecipitation, and found that activated RhoA preferentially bound to Set, which dissociates Set from PP2A. Because Set is a suppressor of PP2A[35,36], the dissociation of Set from PP2A induces PP2A activation and subsequent MAP4K4 inhibition. Conversely, knockdown of Set in RhoA cKO VSMCs canceled the increase in the phosphorylation of MAP4K4 and its downstream MAP kinases, and restored the contractility of VSMCs in the presence of AngII treatment. These results indicate that Set might represent another therapeutic target against AAA. In conclusion, our findings of this study are schematically summarized in Supplementary Fig. 10.

Although research on AAA continues to increase, treatments for this devastating vascular disease are still limited[43,44]. AAA is a life-threatening progressive vascular disease that develops silently and often causes sudden death by aortic rupture. Thus, early diagnosis and the discovery of novel treatments are necessary to inhibit the development of AAA formation. Previous studies have shown that the genetic disorders in VSMCs play crucial roles in AAA[2]. In this regard, the present study adds to the current understanding of AAA pathology, showing that RhoA regulates force generation proteins to maintain aortic contractility and inhibits vascular inflammation, which are mediated by MAP4K4. Although further studies are needed, we speculate that RhoA may be a diagnostic biomarker for AAA and the maintenance of its expression in VSMCs of the aorta is important for counteracting AAA formation and prognosis of this disease.

One of the limitations of this study might be that the AAA formation was induced by the pharmacological stimulation of AngII and BAPN in RhoA cKO mice, while human AAA is generally formed by the result of aging- and/or atherosclerosis-mediated hypertension and aortic wall vulnerability[45,46]. In the basal condition, we could not find differences between control and RhoA cKO mice. There might be a possibility that increased expression of other Rho family members, such as RhoB, compensates for loss of RhoA

in the medial layer of the aorta to keep the aortic structure normal. Given that administration of AngII and BAPN is often used to generate the animal AAA model[47,48], the utilization of these reagents could be acceptable for the investigation of function of the target molecule, such as RhoA, in the process and mechanism of AAA formation. It might be interesting to further examine whether AAA formation is accelerated by aging in RhoA cKO mice. Because it takes too long time and is out of scope of this study, it would be a future experimental research plan.

In conclusion, we successfully showed that VSMC RhoA counteracted the AAA formation in the cKO mouse model, and revealed the mechanism that RhoA suppressed MAP4K4 activity through Set to prevent AAA. As for the translational and clinical implications, a MAP4K4 inhibitor DMX-5804 reversed the adverse phenomena for AAA formation induced by loss of RhoA in VSMCs, providing a therapeutic potential of this inhibitor for AAA in which RhoA expression and MAP4K4 activity were remarkably decreased. This is valuable finding because currently there is no effective pharmaceutical treatment for AAA.

## Methods

**Human aorta collection**. All protocols using human aorta samples were approved by the Research Ethics Committee at Shiga University of Medical Science, and conform to the principles outlined in the Declaration of Helsinki. Aortic tissues were obtained from patients with AAA after surgery. The clinical characteristics of patients were summarized in Supplementary Table 1. All patients provided written informed consent for the use of aortic tissues for this study.

**Generation of VSMC-specific RhoA cKO mice**. We generated RhoA-floxed mice by homologous recombination of the RhoA allele in embryonic stem (ES) cells. The target vector contained exon 3 of the *RhoA* gene flanked by loxP sites and the *neo* gene. ES cells were transfected with the vector and selected by treatment with G418. The *neo* gene was removed by transient administration of Cre recombinase. ES cells harboring the RhoA-floxed allele without *neo* were microinjected into blastocysts to generate chimeric mice. The chimeric mice were further crossed with B6D2F1 mice (CLEA Japan, Tokyo, Japan) to obtain mice heterozygous for the RhoA-floxed allele. The heterozygous mice were backcrossed at least six times onto the C57BL/6 strain, and both male and female heterozygous mice were crossed to obtain mice homozygous for the RhoA-floxed allele. Next, the homozygous mice were mated with transgenic mice expressing the SM22α promoter-driven Cre recombinase (SM22α-Cre) on the C57BL/6 strain, which were purchased from Jackson Laboratory (Bar Harbor, ME, USA). The offspring with both heterozygous RhoA-floxed allele and Cre gene were again mated with the homozygous RhoA-floxed mice to generate VSMC-specific RhoA cKO mice. Littermate mice without the Cre gene were used as a control. Total 106 mice (10–14 weeks old male mice. Control $n = 47$; RhoA cKO $n = 59$) were used in this study. The primers for genotyping to detect mice with Cre and RhoA-floxed allele were as follows: Cre, Forward 5′-AC CTCTGGGAACTGGTCCTT-3′ and Reverse 5′-AGTTACCCCCAGGCTAAGT GC-3′; RhoA-floxed, Forward 5′-TCTTAACCGCTGAGCCATCT-3′ and Reverse 5′-ACCTCTGGGAACTGGTCCTT-3′.

**Animal experiments**. To induce AAA in mice, AngII (1000 ng/kg/min) and BAPN (37.5 mg/kg/d) were administered for 4 weeks using osmotic mini-pumps (ALZET model 1004; Durect Corp., Cupertino, CA, USA) as described previously with some modifications[49]. Mice were anesthetized with isoflurane (1.0–1.5%). Two mini-pumps filled with AngII and BAPN dissolved in sterile saline, respectively, were implanted into the subcutaneous space through a small incision in the back of the

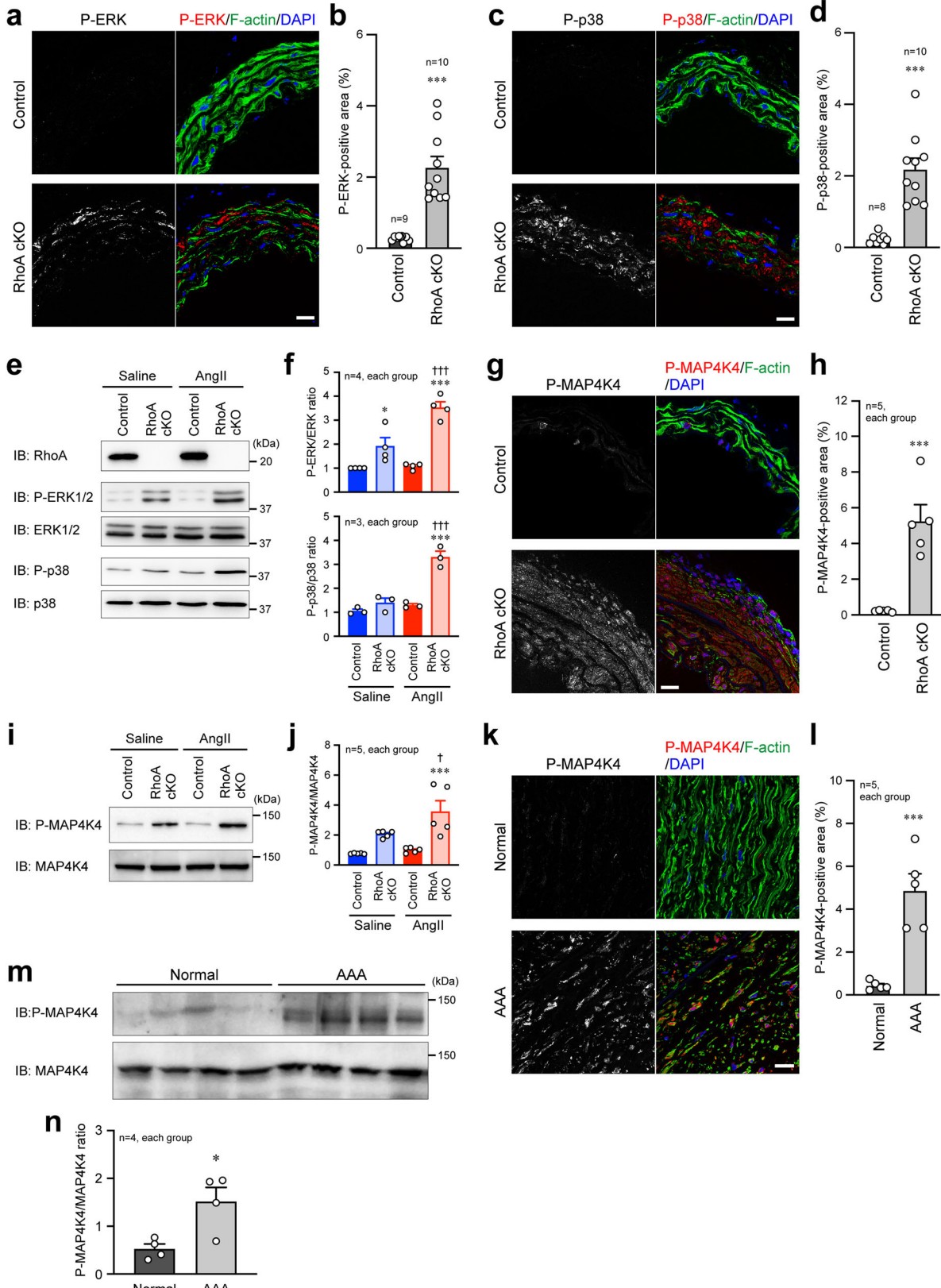

neck. Pumps filled with saline only were used as a control. The incision sites were closed by suture and healed rapidly without any infection. In some mice, the MAP4K4 inhibitor DMX-5804 (2.5 mg/kg) dissolved in 0.5% dimethyl sulfoxide (DMSO) was intravenously administered through the tail vein three times per week for 4 weeks. Aortic diameters and cardiac contractility and dimensions were monitored under isoflurane (1.0–1.5%) anesthetization by ultrasonography (Vevo 2100; VisualSonics Inc., Toronto, Canada). All animal experiments were approved

by Shiga University of Medical Science Animal Care and Use Committee and were performed in accordance with relevant guidelines and regulations including Animal Research Reporting of In Vivo Experiments (ARRIVE) guidelines.

**Measurement of blood pressure**. Arterial BP was measured weekly in conscious mice using the plethysmographic tail-cuff method (model BP-98-AL; Softron,

**Fig. 5 Activation of MAP kinase signaling molecules in VSMCs by loss of RhoA. a**, **c**, **g**, **k** Immunofluorescence images of control and RhoA cKO mouse aorta after the 4-week treatment with AngII+BAPN (**a**, **c**, **g**) and of normal and AAA areas in human aorta (**k**). F-actin and nuclei were visualized with phalloidin and DAPI, respectively. **b**, **d**, **h**, **l** Summary graph of the percentage of positive area for each MAP kinase signaling molecule. $n = 9$ in Control and $n = 10$ in RhoA cKO (**b**), $n = 8$ in Control and $n = 10$ in RhoA cKO (**d**), and $n = 5$, each group (**h**, **l**). **e**, **i** Western blotting with the indicated antibodies using cultured aortic VSMCs after treatment with saline or AngII for 24 h. GAPDH served as the loading control. **f**, **j** Summary graph of the phosphorylated/total MAP kinase signaling molecule ratio in each group. $n = 3$–$5$, each group. **m** Western blotting for P-MAP4K4 and total MAP4K4 in normal and AAA areas in human aorta. **n** Summary graph of the P-MAP4K4/MAP4K4 ratio. $n = 4$, each group. Scale bars: 20 µm (**a**, **c**, **g**, **k**). In (**b**, **d**, **l**, **n**), the data between the two groups were analyzed by $t$-test, and in (**f**, **j**), one-way ANOVA was applied to compare the data between groups. *$p < 0.05$ and ***$p < 0.001$ vs. Control or Normal; †$p < 0.05$ and †††$p < 0.001$ vs. RhoA cKO in the saline treatment.

Tokyo, Japan). Mice were warmed for at least 5–10 min at 37 °C in the cylindrical thermostat before and during the course of BP measurement. BP was measured in 2-min intervals, and the mean of seven steady-state measurements was used as the BP in each mouse[50].

**Histological staining**. Mice were euthanized by cervical dislocation, and the aorta was perfused sequentially with cold phosphate-buffered saline (PBS) and 10% phosphate-buffered formalin at physiological pressure for 5 min each time. The extracted mouse and human aorta samples were fixed with 10% phosphate-buffered formalin for 24 h and embedded in paraffin. Cross-sections (4 µm thick) were prepared and deparaffinized. H-E staining was performed to detect the morphology of the aortas. For Verhoeff Van Gieson staining, formalin-fixed and deparaffinized sections were incubated with Verhoeff's solution for 1 h, counterstained with Van Gieson's solution, quickly dehydrated, and mounted on the glass slide. Degradation of medial elastic lamina was analyzed by grading of degradation (no degradation, mild degradation, severe degradation, and aortic rupture)[51].

**Antibodies**. Detailed information of the primary antibodies used in this study was summarized in Supplementary Table 2.

**Immunohistochemistry**. Formalin-fixed paraffin-embedded human aorta samples were sectioned and deparaffinized. The samples were soaked in pre-heated antigen retrieval solution (1 mmol/L citric acid, 0.05% Tween 20 at pH 6) at 90–100 °C for 20–30 min, and cooled down at room temperature. Endogenous peroxidase activity in all sections was quenched using 3% hydrogen peroxidase, and samples were then blocked with 3% bovine serum albumin (BSA) prior to overnight incubation with an anti-RhoA antibody. After washing with PBS, the samples were incubated with a biotin-conjugated secondary antibody (1:200 dilution; Thermo Fisher Scientific, Waltham, MA, USA) for 1 h at room temperature, followed by incubation with streptavidin-peroxidase (Nacalai Tesque, Kyoto, Japan) for 30 min. The avidin–biotin complex was detected with the Strep ABC peroxidase kit (Nacalai Tesque). Samples were counterstained with hematoxylin, followed by dehydration steps.

**Immunofluorescence staining**. Cryosections of human and mouse aorta samples, and mouse adult heart samples were fixed with 4% paraformaldehyde (PFA) for 10 min, and then, washed three times with PBS for 5 min each time. The samples were permeabilized with 0.2% Triton X-100 in PBS for 30 min and blocked with 3–5% BSA for 30 min at room temperature to avoid nonspecific staining, followed by incubation with the primary antibody overnight at 4 °C. The fluorescent dye-labeled secondary antibody (Alexa Fluor 488 and Alexa Fluor 555, 1:200–1000 dilution, Thermo Fisher Scientific), rhodamine phalloidin (1:600 dilution, Thermo Fisher Scientific), and Alexa Fluor 488/647 phalloidin (1:600 dilution, Thermo Fisher Scientific) were applied for 1 h at room temperature. The samples were washed three times with PBS for 5 min each time, and then, incubated with mounting solution with DAPI for nuclei staining (Dojindo, Kumamoto, Japan). Immunofluorescence images were taken by a confocal microscope (FV-1000; Olympus, Tokyo, Japan, or Leica SP8; Leica Microsystems, Wetzlar, Germany).

For staining of mouse embryos, the embryos at E11.5 were fixed with 100% methanol overnight at 4 °C. The hearts were extracted and washed with 50% methanol in PBS containing 0.5% Triton X-100 (PBST) for 30 min at 4 °C, followed by washing with PBST for 30 min at 4 °C. The samples were blocked with 1% milk in PBST for 20 min at 4 °C, and were incubated with the primary antibodies for RhoA and α-myosin heavy chain (α-MHC) for 48 h at 4 °C. Then, they were washed three times with PBST, and were incubated with the fluorescent dye-labeled secondary antibodies for 48 h at 4 °C. Finally, they were washed three times with PBST, and incubated with benzyl-alcohol:benzyl-benzoate (2:3) solution for optical clearing. Immunofluorescence images were taken by Leica SP8 confocal microscope.

To validate the specificity of antibodies used for immunostaining of phosphorylated (activated) MAP kinase signaling molecules, the following inhibitors were administered into RhoA cKO mice after the 4-week treatment with AngII and BAPN: PD-98059 for the ERK inhibitor, SB-203580 for the p38

inhibitor, and DMX-5804 for the MAP4K4 inhibitor. Each inhibitor was dissolved in 0.5% DMSO, and 100 µL of the inhibitor solution (PD-98059: 10 mg/kg, SB-203580: 5 mg/kg, and DMX-5804: 2.5 mg/kg) was intravenously injected through the tail vein of RhoA cKO mice. The solvent of 0.5% DMSO was intravenously injected into other mice as a vehicle. At 30 min after the injection, the aorta was harvested and the frozen sections were prepared for immunostaining.

**RNA isolation and real-time PCR**. Total RNA was isolated from human and mouse aortas and mouse VSMCs using TRIzol RNA isolation reagent (Thermo Fisher Scientific). cDNA was synthesized by the ReverTra Ace qPCR RT Master Mix with gDNA Remover (Toyobo, Osaka, Japan). After reverse transcription, quantitative real-time PCR was performed using the LightCycler Instrument (Roche Diagnostics, Basel, Switzerland). The real-time PCR data were quantified by the standard curve method and human or mouse GAPDH mRNA expression level served as an internal control. The primers were shown in Supplementary Table 3.

**Plasmid construction**. The expression plasmid of EGFP-tagged RhoA-WT was prepared previously[52]. cDNAs of mouse Set isoform 1 (Set1) and isoform 2 (Set2) were obtained by reverse transcription PCR using RNA extracted from mouse aortic VSMCs. The primers for amplifying the cDNAs were as follows: Set1, Forward 5′-GGGGAATTCGCCCCGAAGCGGCAGTCTGCGAT-3′ and Reverse 5′-GGGCTCGAGCTAATCATCCTCGCCTTCGTCCTCCT-3′; Set2, Forward 5′-GGGGAATTCTCTGCGCCGACGGCCAAAGCCAG-3′ and Reverse 5′-GGGCTCGAGCTAATCATCCTCGCCTTCGTCCTCCT-3′. Each cDNA was subcloned into the pCMV-HA expression plasmid (Takara Bio, Kusatsu, Japan) by digesting them with EcoRI and XhoI restriction enzymes and ligating them with Ligation High Ver.2 (Toyobo, Osaka, Japan). The sequences of cDNAs inserted into the plasmid were confirmed by Prism 3100xl Genetic Analyzer (Applied Biosystems, Waltham, MA, USA).

**Western blotting**. Human and mouse aortas, and mouse hearts were homogenized mechanically in radioimmunoprecipitation assay (RIPA) buffer (50 mmol/L Tris-HCl [pH 7.5], 150 mmol/L NaCl, 0.5% sodium deoxycholate, 0.1% sodium dodecyl sulfate (SDS), 1% Nonidet P-40, 1 µg/mL aprotinin, 1 µg/mL leupeptin, 1 mmol/L phenylmethylsulfonyl fluoride, 5 mmol/L sodium fluoride, and 1 mmol/L sodium orthovanadate). VSMCs were also lysed in RIPA buffer. The lysates were centrifuged at 14,000 rpm for 15 min, and the supernatant was used for further experiments. Protein samples were separated by SDS-polyacrylamide gel electrophoresis (PAGE) and transferred to a polyvinylidene difluoride membrane (Bio-Rad Laboratories, Hercules, CA, USA). The membrane was then blocked for 1 h at room temperature in 5% BSA or 5% skim milk in Tris-buffered saline with Tween 20. The membrane was incubated with primary antibody overnight in 5% skim milk at 4 °C, followed by incubation with horseradish peroxidase (HRP)-labeled secondary antibody (GE Healthcare, Piscataway, NJ, USA) for 1 h in 5% skim milk. The membrane was incubated with HRP substrate (Luminata Forte, Millipore Corp., Billerica, MA, USA) for 5 min and observed on a luminescent image analyzer LAS-4000 (Fujifilm Life Science, Tokyo, Japan). Band densities were analyzed using ImageJ software.

**Measurement of collagen synthesis and cross-linked collagen**. Snap frozen aortic samples from control and RhoA cKO mice were pulverized and washed once with PBS for 30 min at 4 °C. An aliquot of the sample was transferred into a new tube without pepsin digestion to determine the total collagen content for evaluating collagen synthesis. After centrifugation at $16,000 \times g$ for 30 min at 4 °C, the precipitated samples were digested in 0.5 mol/L of acetic acid containing 1 mg/mL of pepsin for 16 h at 4 °C with gentle rotation. The undigested samples were assembled by centrifugation at $16,000 \times g$ for 30 min at 4 °C, and were collected as cross-linked collagen. The supernatant was collected as non-cross-linked collagen, and was subsequently precipitated in 2 mol/L of NaCl for 30 min, followed by centrifugation at $16,000 \times g$ for 30 min at 4 °C. Total, cross-linked and non-cross-linked collagens were quantified by the hydroxyproline assay[53], which were performed using the kit (Sigma-Aldrich, St. Louis,

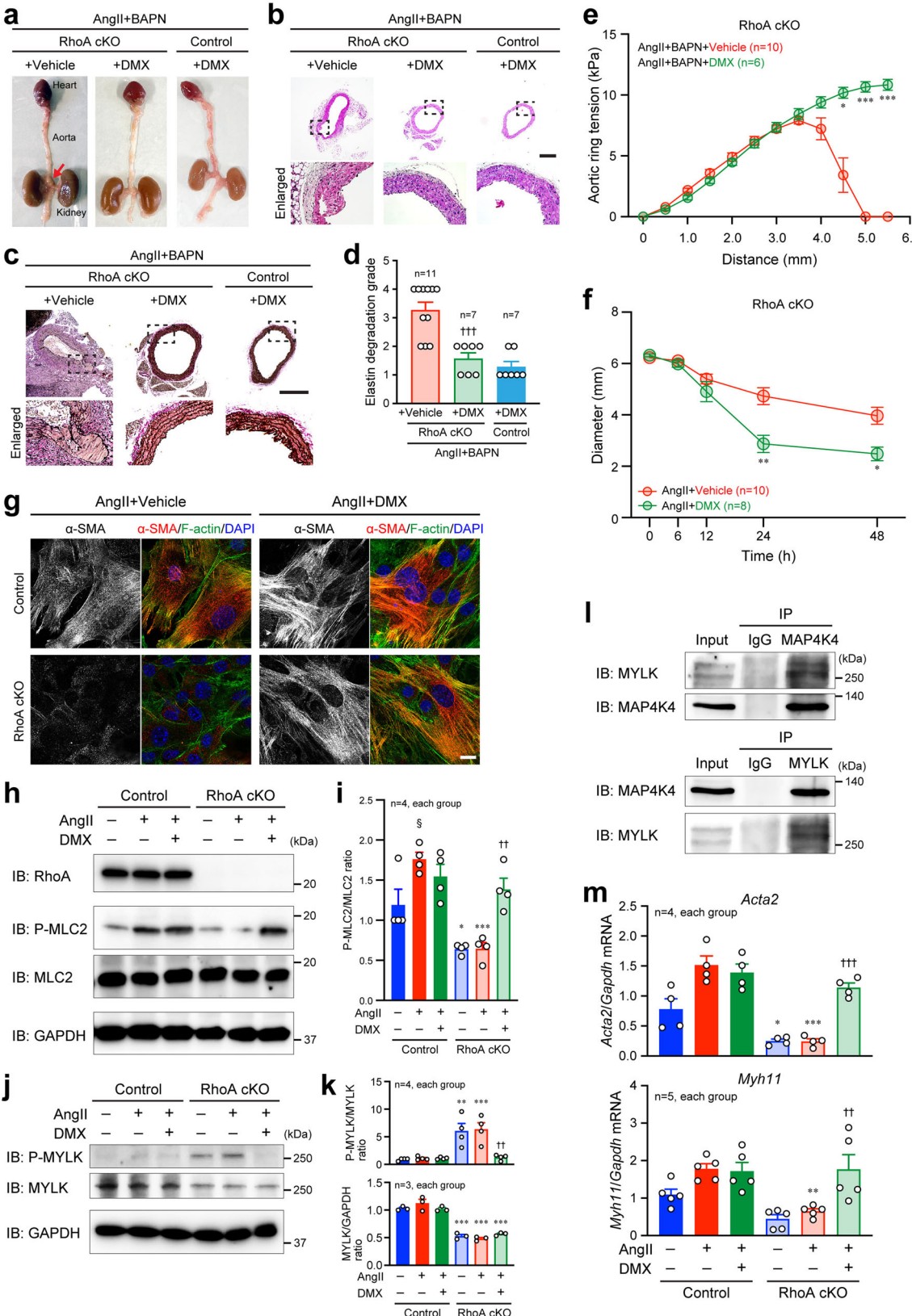

MO, USA). Results are reported as micrograms of hydroxyproline per milligram of wet tissue weight.

**Harvest of mouse aortic VSMCs and plasmid introduction.** To isolate mouse primary aortic VSMCs, the whole aorta from RhoA cKO and control mice was

extracted, cut into small pieces, and incubated with Dulbecco's modified Eagle's medium (DMEM) containing collagenase type I (Sigma-Aldrich), elastase (Sigma-Aldrich) and trypsin inhibitor (Thermo Fisher Scientific) for 3–4 h at 37 °C in a humidified atmosphere of 5% $CO_2$ and 95% air[54]. VSMCs were cultured in DMEM containing 10% FBS. VSMCs of passages 4 to 7 at 70%–80% confluence were used for experiments. The characteristics of the VSMC populations were confirmed by

**Fig. 6 Rescue of RhoA knockout-mediated adverse phenotypes by treatment with a MAP4K4 kinase inhibitor DMX-5804. a** External appearance of the aorta with the heart and kidneys extracted after the 4-week treatment with AngII and BAPN in the presence of 0.5% DMSO (Vehicle) or DMX-5804 (DMX). An arrow indicates formed AAA. **b** H-E staining of the aorta. **c** Verhoeff Van Gieson staining of the aorta. **d** Summary graph of the elastin degradation grade analyzed in (**c**). $n = 7–11$, each group. **e** Changes of aortic ring tension measured by wire myograph. $n = 10$ in AngII+BAPN + Vehicle and $n = 6$ in AngII+BAPN + DMX. **f** Changes of aortic VSMCs-containing collagen gel diameter in in vitro collagen gel contractility assay. $n = 10$ in AngII +Vehicle and $n = 8$ in AngII+DMX. **g** Immunofluorescence images of aortic VSMCs stained with the α-SMA antibody after treatment with AngII in the presence of vehicle or DMX-5804 for 24 h. F-actin and nuclei were visualized with phalloidin and DAPI, respectively. **h, j** Western blotting with the indicated antibodies using aortic VSMCs after treatment with or without AngII or DMX-5804 for 24 h. GAPDH served as the loading control. **i, k** Summary graphs of the P-MLC2/MLC2, P-MYLK/MYLK, and MYLK/GAPDH ratios. $n = 4$ (**i**), $n = 3–4$ (**k**), each group. **l** Immunoprecipitation experiments using the indicated antibodies. IgG was used as the negative control for immunoprecipitation. **m** qPCR analysis using VSMCs treated with or without AngII and/or DMX-5804 for 24 h. $n = 4–5$, each group. Scale bars: 500 μm (**b, c**) and 20 μm (**g**). One-way (**d, i, k, m**) or two-way (**e, f**) ANOVA was applied to compare the data between groups. *$p < 0.05$, **$p < 0.01$ and ***$p < 0.001$ vs. Vehicle or Control; ††$p < 0.01$ and †††$p < 0.001$ vs. AngII treatment without DMX-5804; §$p < 0.05$ vs. No treatment.

immunofluorescent staining for α-SMA (Santa Cruz Biotechnology). For in vitro experiments, the cells were treated with saline, 1 μmol/L AngII, and/or 10 μmol/L DMX-5804 (Selleck, Houston, TX, USA).

Plasmid DNA was introduced into VSMCs by the Neon Transfection System (Invitrogen, Carlsbad, CA, USA). After washing the cells with PBS, the cells ($1 × 10^7$ cells/100 μL) were resuspended in Buffer R and mixed with 500 ng of each plasmid. The plasmid introduction parameters were as follows: pulse voltage 1200 V, pulse width 20 ms, and pulse number 2.

**Aortic tension measurement**. The isolated mouse aorta was cut in round slices (~2 mm in width), and mounted to the wire of a myograph (Multi Wire Myograph System-Model 620 M; Danish Myo Technology, Aarhus, Denmark) in a bath filled with Krebs' solution (118 mmol/L NaCl, 25 mmol/L NaHCO₃, 11 mmol/L glucose, 4.7 mmol/L KCl, 1.17 mmol/L MgSO₄, 1.2 mmol/L KH₂PO₄, 2.5 mmol/L CaCl₂ at 37 °C), continuously bubbled with 95% $O_2$ and 5% $CO_2$. The aorta was equilibrated to the quiescent state for 5 min in the condition. The mouse aortic tension was measured using the Multi Wire Myograph System[55]. The obtained pressures were monitored and recorded using the LabChart 8 software (ADInstruments Inc., Colorado Springs, CO, USA).

**Collagen gel contractility assay**. Type I collagen solution derived from porcine tendon (Nitta Gelatin, Osaka, Japan), DMEM (twice concentrated), reconstituting buffer (50 mmol/L NaOH, 260 mmol/L NaHCO₃, 200 mmol/L HEPES), and VSMCs suspended in DMEM were mixed on ice at ratio of 7:2:1:1. Next, one drop (30–50 μL) of the mixed solution ($4 × 10^5$ cells/mL) was placed on a siliconized cover slip in a 6-well plate. The plate was incubated at 37 °C for 30 min to harden the gel, and 2 mL of DMEM containing 10% FBS with AngII (final concentration 10 μmol/L) or saline was added onto the gel. The cells were cultured for 24 h at 37 °C in a humidified atmosphere of 5% $CO_2$ and 95% air. After the culture, the medium was removed from each well, and the gel was carefully detached from the well. Then, the gel diameter was measured at 0, 6, 12, 24, and 48 h to measure the contractility of isolated VSMCs[56].

**Vascular permeability assay**. Vascular permeability associated with the endothelial barrier function in the aorta was determined by the procedure using Evans Blue dye[57]. One hundred μL of 1% Evans Blue dye (Nacalai Tesque) dissolved in saline was intravenously injected into mice. At 10 min after the injection, the mice were euthanized and aorta was harvested, followed by fixation with 10% formalin solution. The aorta was observed under light microscope to take pictures. The area stained with Evans Blue dye was measured to evaluate the damage of the endothelial layer.

**Gelatin zymography**. Lysed human aorta samples were mixed with 5x non-reducing sample buffer (312.5 mmol/L Tris-HCl [pH 6.8], 11.5% SDS, 40% glycerol), and loaded onto a 7.5% SDS-PAGE gel containing 4 mg/mL gelatin. After SDS-PAGE, the gel was washed twice with washing buffer (50 mmol/L Tris-HCl [pH 7.5], 2.5% Triton X-100, 5 mmol/L CaCl₂, 1 μmol/L ZnCl₂) at 37 °C for 30 min each time, and then incubated in incubation buffer (50 mmol/L Tris-HCl [pH 7.5], 1% Triton X-100, 5 mmol/L CaCl₂, 1 μmol/L ZnCl₂) for 10 min with gentle agitation, followed by replacement with fresh incubation buffer and incubation for 24 h at 37 °C. After Coomassie Brilliant Blue (CBB; Nacalai Tesque) staining, areas of gelatin degradation were visible as clear and sharp bands against a blue background of the non-degraded substrate. Band densities were analyzed using ImageJ software (National Institute of Health, Bethesda, MD, USA).

**Immunoprecipitation and LC-MS/MS**. The lysates of aortic VSMCs were pre-cleared with protein G Sepharose beads (GE Healthcare) to remove proteins that non-specifically bind to the beads. After centrifugation, the supernatants were incubated with the indicated primary antibody (1:100 dilution) overnight at 4 °C,

followed by incubation with protein G Sepharose beads for 2 h at 4 °C. Samples were centrifuged at $3000 × g$ for 1 min at 4 °C, and the beads were washed three times with RIPA buffer. Proteins bound to the beads were eluted in 2x Sample loading buffer, and used in western blotting.

For LC-MS/MS analysis, the gel was stained with CBB overnight, and the specific protein bands in the GFP-RhoA-WT lane were excised after destaining. The gel slices were soaked in 50% acetonitrile and 50 mmol/L ammonium bicarbonate solution for complete destaining of CBB. After washing with deionized water, the slices were soaked in 100% acetonitrile for 15 min and dried completely using Savant SpeedVac (Thermo Fisher Scientific). Proteins in the gel slices were digested with 800 ng/40 μL trypsin: lysyl endopeptidase mix (1:1) at 37 °C for 16 h. The digested peptide samples were transferred to a new tube, and desalted using C18-StageTip (Thermo Fisher Scientific). After the samples were dried with Savant SpeedVac, 10 μL of 0.1% formic acid/3% acetonitrile solution was added to solubilize the peptides for the next application.

For LC, 2 μL of the sample was applied to the Nanoflow UPLC (UltiMate 3000 RSLCnano LC System, Thermo Fisher Scientific) equipped with a Nanocolumn (75 μm × 125 mm column filled with a reversed-phase ReproSil-Pur C18-AQ resin; Nikkyo Technos Co., Ltd, Tokyo, Japan) at 40 °C. The gradient elution program with mobile phase A (0.1% formic acid in water) and mobile phase B (0.1% formic acid in 80% acetonitrile) was performed for separation of absorbed peptides: phase B, 2%–8% for 0–4 min, 8%–30% for 4–26 min, 30%–65% for 26–34 min. Subsequently, tandem mass spectrometry (MS/MS) was carried out using Orbitrap Q Exactive HF-X mass spectrometry (Thermo Fisher Scientific). The electrospray ionization source was operated with positive ion mode, voltage set to 1.7 kV and ion transfer capillary set to 275 °C. MS1 spectra were collected in the range of 380 to 1240 $m/z$ at 60,000 resolution to hit an AGC target of $3 × 10^6$. The top 40 precursor ions with charge states of 2+ to 5+ were selected for fragmentation with stepped normalized collision energy of 22%, 26%, and 30%. MS2 spectra were collected at 15,000 resolution to hit an AGC target of $1 × 10^5$. The dynamic exclusion time was set to 12 s. The raw data processing and analysis by database searches were performed with PEAKS STUDIO Desktop Version X + referring 17,023 of protein-entries in the mouse UniProtKB/Swiss-Prot mouse database (UniProt id UP000000589, reviewed, canonical; downloaded on February 10, 2020). The parameters of the software were set as follows: oxidation (+15.99 Da) (methionine residues) was set as considered variable modifications of proteins; trypsin-cleavages were permitted; the precursor ion mass tolerance was set to ±10.0 ppm, and MS/MS tolerance was set to 0.015 Da; the protein identification threshold was set at <1% for both peptide and protein false discovery rates. To estimate the false discovery rate in the PEAKS studio software, the decoy fusion method was applied. The proteome analysis was conducted by Kazusa DNA Research Institute (Kisarazu, Japan). The PSOPIA software (National Institute of Biomedical Innovation, Health and Nutrition, Ibaraki, Japan) was used for in silico analysis to identify the protein(s) that can interact with components of the STRIPAK complex, which modulates MAP4K4 activity.

**RhoA pull-down assay**. Active GTP-bound RhoA was detected by pull-down assay as described previously with some modifications[52]. Briefly, cells were lysed in the cell lysis buffer (50 mmol/L Tris/HCl [pH 7.5], 0.5 mol/L NaCl, 10 mmol/L MgCl₂, 1% Triton X-100). The cell extract was obtained by centrifugation, and incubated with glutathione S-transferase fused with the mDia Rho-binding domain conjugated with glutathione sepharose beads (GE Healthcare) at 4 °C overnight to collect GTP-bound RhoA. After the beads were washed with the cell lysis buffer, proteins bound to the beads were eluted with SDS sample buffer and subjected to SDS-PAGE, followed by western blotting with anti-RhoA antibody. The active GTP-interacting protein Set was also detected by an anti-Set antibody.

**siRNA transfection**. Set siRNA and negative control (Scramble) RNA were produced using the CUGA7 in vitro transcription kit (Nippon Gene, Tokyo, Japan). The siRNA sequences are as follows: Set siRNA 5′-

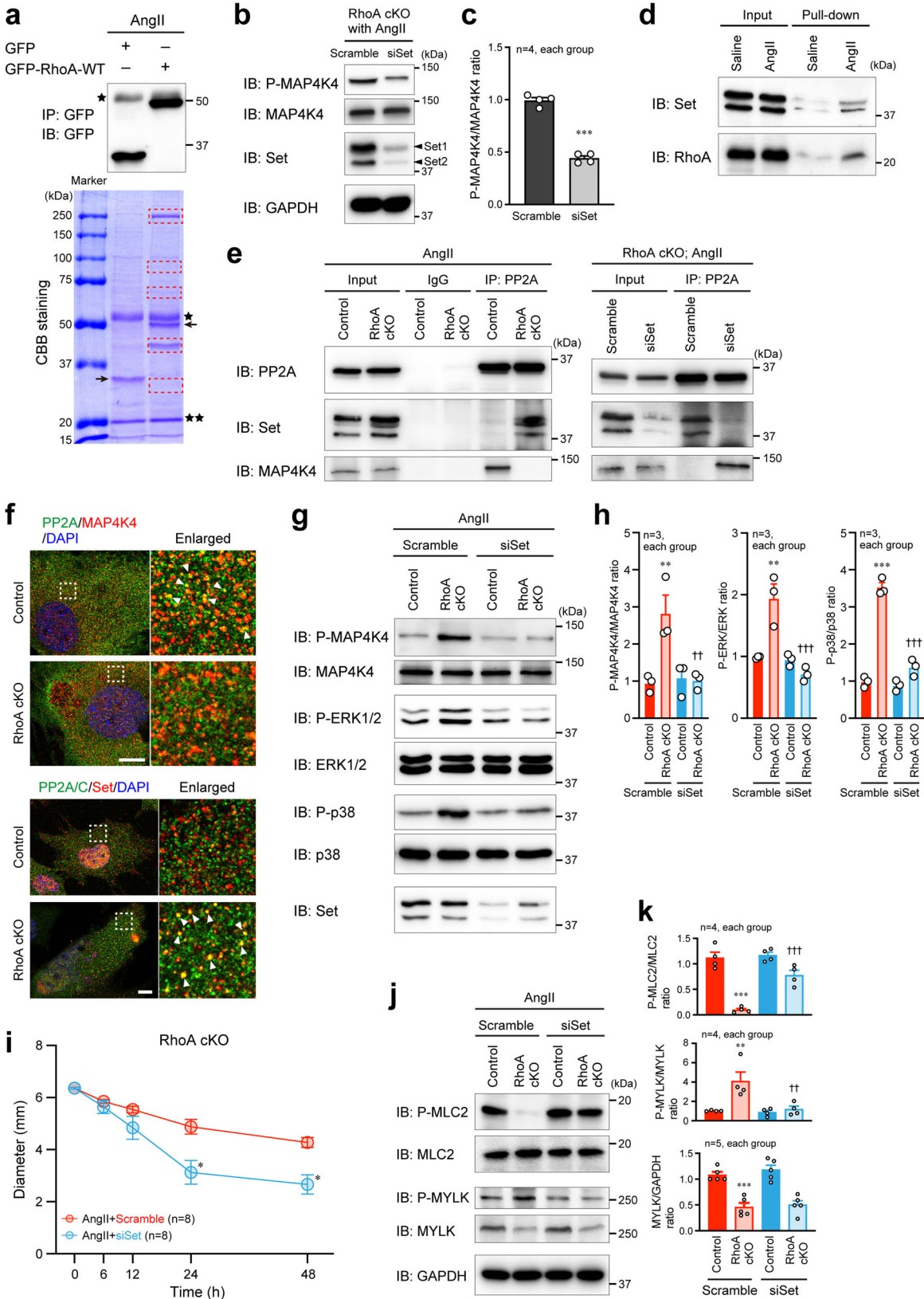

GGATGAAGGTGAAGAAGAT-3′ and Scramble 5′-CAGTCGCGTTTGC-GACTGG-3′. siRNA transfection was performed using Lipofectamine RNAiMAX transfection reagent (Invitrogen).

**Statistics and reproducibility**. All data are expressed as the mean ± standard error of the mean (SEM). All experiments were performed at least three times

independently. Statistical analysis was performed using Prism 9 software (Graph-Pad Software, La Jolla, CA, USA). Statistical differences between experimental groups were evaluated by the Fisher's exact test, two-tailed Student's $t$-test, one-way analysis of variance (ANOVA), or two-way repeated measures ANOVA. If ANOVA indicated overall significance, individual differences were evaluated using the Bonferroni's post-test. $p < 0.05$ was considered to be statistically significant.

**Fig. 7 Identification of the molecule that mediates the signaling between RhoA and MAP4K4. a** After transfection of GFP or GFP-RhoA-WT expression plasmid, VSMCs were stimulated with AngII for 24 h. The cell extracts were immunoprecipitated and immunoblotted with the anti-GFP antibody (upper panel), and the immunoprecipitated proteins to the anti-GFP antibody were detected by Coomassie Brilliant Blue (CBB) staining (lower panel). Single star and double stars: IgG heavy and light chains, respectively. Arrows: transfected GFP or GFP-RhoA-WT. Bands surrounded by red dotted rectangles were further analyzed by LC-MS/MS. **b** Western blotting for P-MAP4K4 and MAP4K4 in aortic RhoA cKO VSMCs transfected with scramble or siSet siRNA. **c** Summary graph of the P-MAP4K4/MAP4K4 ratio. **d** Cell extracts from control VSMCs after saline or AngII treatment were used in pull-down assays with GST fusion protein including the GTP-bound RhoA-binding domain of mDia, followed by western blotting with anti-Set or anti-RhoA antibody. **e** After stimulation of control or RhoA cKO VSMCs with AngII for 24 h, cell extracts were immunoprecipitated with anti-PP2A antibody or control IgG, followed by western blotting with the indicated antibodies. **f** Immunofluorescence images of control or RhoA cKO VSMCs stained with the indicated antibodies after treatment with AngII for 24 h. Nuclei were visualized with DAPI. Scale bars: 20 μm. **g**, **j** After transfection of cells with scramble or siSet siRNA and AngII stimulation for 24 h, cell extracts of control or RhoA cKO VSMCs were immunoblotted with the indicated antibodies. **h**, **k** Summary graphs of the phosphorylated/total MAP kinase signaling molecule ratios (**h**) and P-MLC2/MLC2, P-MYLK/MYLK, and MYLK/GAPDH ratios (**k**). $n = 3$ (**h**) and $n = 4–5$ (**k**), each group. **i** Summary graph of VSMC-containing in vitro collagen gel contractility assay using AngII-stimulated RhoA cKO VSMCs transfected with scramble or siSet siRNA. $n = 8$, each group. The data were analyzed by $t$-test (**c**), one-way ANOVA (**h**, **k**) or two-way ANOVA (**i**) to compare the groups. In (**c**), $***p < 0.001$ vs. Scramble; in (**h**, **k**), $**p < 0.01$ and $***p < 0.001$ vs. Control, $††p < 0.01$ and $†††p < 0.001$ vs. Scramble; in (**i**), $*p < 0.05$ vs. Scramble.

**Reporting summary**. Further information on research design is available in the Nature Research Reporting Summary linked to this article.

## Data availability

Uncropped western blot images are shown in Supplementary Figs. 11–14, and all source data underlying the graphs in the main figures are shown in Supplementary Data 2. Newly generated plasmids in this study are deposited in Addgene (https://www.addgene.org/) with the deposition ID number 81677. Mass spectrometry-based proteomics data are deposited to the ProteomeXchange Consortium (http://proteomecentral.proteomexchange.org) via the PRIDE partner repository with the dataset identifier PXD036825[58]. All other datasets generated during and/or analyzed during this study are available from the corresponding author on reasonable request.

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

## Acknowledgements

We thank Mr. Takefumi Yamamoto at the Central Research Laboratory, and Prof. Eiichiro Nishi at Department of Pharmacology, Shiga University of Medical Science for their excellent technical assistance in this study. This study was supported in part by Grants-in-aid for Scientific Research <KAKENHI> from Japan Society for the Promotion of Science for A.Shimizu [21K06854], M.K. [21K16086], A.Sato [21K09419], and H.O. [17K08627 and 20K08489]; Takeda Science Foundation, The Naito Foundation, SENSHIN Medical Research Foundation, and Uehara Memorial Foundation for H.O.

## Author contributions

Designing research studies (M.R.M., A.Shimizu, and H.O.), conducting experiments and acquiring data (M.R.M., A.Shimizu, M.K., N.I.A.R., J.E.C.S., L.K.C.N., M.R.K., W.W.T., S.C., X.P., M.T-O., and A.Sato), analyzing data (M.R.M., A.Shimizu, N.T., T.S., and H.O.), providing samples (M.T-O., N.T., and T.S.), and writing the manuscript (M.R.M., A.Shimizu, and H.O.).

## Competing interests

The authors declare no competing interests.
