## [Peer Review File · Communications Biology]

Reviewers' comments:

Reviewer #1 (Remarks to the Author):

In this manuscript, the authors showed comprehensive data on the protective effects of vascular smooth muscle RhoA in the progress of AAA formation. They determined that activated RhoA bound to Set, a suppressor of PP2A, and dissociated Set from PP2A and thus induced PP2A activation as well as subsequent MAP4K4 inhibition. MAP kinase cascade activation induced by MAP4K4 were therefore blocked and vascular inflammation was impeded, while vascular contractility was promoted and finally the AAA pathology was inhibited. They also proposed that RhoA may serve as a diagnostic biomarker for AAA. Overall, the data could support the final conclusion. But the manuscript needs to be strengthened in some areas.

1. In the manuscript, it seems to be insufficient to conclude that 'RhoA in VSMCs has a protective role against the stimulation to form AAA' only through a series of experiments in Figure 2 without any remedy experimental data in animal or VSMC level.

2. In Figure 1A, the images of normal and AAA lesions in the human aorta did not seem to be in the same scale.

3. Were there any differences in the length of aorta between control and RhoA cKO mice as shown in Figure 2B?

4. How to explain the phenomenon that AngII treatment increased the expression of α -SMA and F-actin (as shown in Figure 3E-H) but induced AAA both in control and RhoA cKO mice (as shown in Table 1)?

5. There are a little grammatical mistakes in the manuscript, please check and correct them.

Reviewer #2 (Remarks to the Author):

Hisakazu Ogita

Title: Vascular Smooth Muscle RhoA Counteracts Abdominal Aortic Aneurysm Formation by Modulating Mitogen-Activated Protein Kinase Kinase Kinase Kinase 4 Activity

This manuscript further underscores the importance of RhoA signaling in vascular smooth muscle cell function and the development of abdominal aortic aneurysm in mice and man. Moreover, the data support a role for MAP4K4 in mediating RhoA-dependent effects and provide evidence for a new molecular player that links these signaling pathways. In general the *in vivo* data provide strong support for the role of the RhoA-MAP4K4 signaling axis in the control of vessel remodeling and AAA formation, however the link between RhoA and MAP4K4 is less well developed.

Specific Comments:

Fig 1. The studies in figure 1 nicely demonstrate that a reduction in RhoA levels is strongly associated with the development of AAA in patients and this correlates with upregulated TIMP/MMP levels (Fig S2). Is there also an association with phosphorylated MAP4K4? It would also be more compelling if some smooth muscle, fibroblast and EC marker levels were shown in the immunoblot (panel H) to get a better feel for the cell types in each extract.

Fig 2 and Fig S1. It is somewhat surprising that depletion of RhoA in SMC compartments from birth has little impact on basal vessel growth and function. While the data provided generally support this conclusion, the statement that blood pressure is not impacted is overstated given that this was performed as a spot check using a tail cuff method. This conclusion should be softened since this method is not sufficient to assess dynamic changes in BP, particularly during the active period- which would require continuous monitoring by radio-telemetry. Also, since the SM22-Cre model also leads to gene inactivation in the developing heart, this should be taken into consideration (see below).

Nonetheless, since the basal characteristics shown do not change, it would be of interest to explore the extent to which RhoB and C are upregulated to compensate for losses of RhoA. Moreover, given the choice of AAA induction that includes application of the lysyl oxidase inhibitor, studies should be done to determine if there are baseline changes in collagen production and collagen cross-linking in the RhoA knockout vessels.

Figure 4. The conclusion that SMC-restricted depletion of RhoA leads to reduced endothelial barrier is overstated. Further studies using dye accumulation to assess barrier function would be required to support this conclusion. Was disruption observed all along the AA or simply at the locale of the

aneurism?

Figure 5. The staining for phosphorylated protein levels in vessel sections is not particularly convincing and would require some control validation studies showing that the signal detected was specific (i.e. reduced following inhibitor treatment or in validated knock-outs).

Figure 6. The rescue studies with DMX-5084 are impressive, however controls are missing to show the impact of DMX-5084 on WT treated mice. This is critical to support the conclusions drawn. Also with respect to the comment above that SM22-Cre would be expected to reduce RhoA levels in the heart and prior studies showing that DMX-5084 impacts cardiac contractility, further studies are needed to test this putative interaction in AAA development/progression. Finally, were there any differences in plasma volumes between the treated and un-treated mice of the various genotypes?

Figure 7. The mechanistic studies to reveal a direct link between RhoA-SET-MAP4K4-MLCp are too preliminary to support the conclusions drawn. While studies from this manuscript confirm that MAP4K4 is downstream of RhoA – the idea that SET is a major link between active RhoA and MAP4K4 requires further study. Other groups have convincingly shown that RhoA links to MAP4K4 through the well-known RhoA binding partner Rhophilin (Nature Cell Biology volume 21, pages 1565–1577 (2019). Also, SET has previously been shown to be an effector of Rac1 (Oncotarget. 2017 Sep 15; 8(40): 67966–67979). These findings should be integrated into the current studies. Many additional biochemical studies would be necessary to validate SET as a RhoA effector.

Response to Reviewers' Comments:

Before responding to the reviewers' comments, we greatly thank each reviewer for his/her critical reading of our manuscript and providing us with valuable suggestions to improve the manuscript. The detailed responses to the reviewers' comments are described below.

Reviewer #1

1. In the manuscript, it seems to be insufficient to conclude that 'RhoA in VSMCs has a protective role against the stimulation to form AAA' only through a series of experiments in Figure 2 without any remedy experimental data in animal or VSMC level.

As the reviewer's constructive suggestion, the expression plasmid for GFP-tagged RhoA wild-type (WT) was transfected in RhoA cKO VSMCs to confirm that the abnormalities observed in RhoA cKO VSMCs were due to loss of RhoA in the cells. We found that the reduced expression of α -SMA and other VSMC contractile marker genes was recovered, and that disorganized F-actin staining was normalized by the RhoA re-expression. Moreover, the increase in the expression of inflammatory cytokines in RhoA cKO VSMCs was reversed by the transfection of GFP-RhoA-WT in the cells, and AngII-induced enhancement of MAP kinase signaling pathway in RhoA cKO VSMCs was returned to the level in control VSMCs by re-expression of RhoA. The results of these remedy experiments were shown in Supplementary Fig. 4 and described on page 9, lines 3–10 in the revised manuscript.

2. In Figure 1A, the images of normal and AAA lesions in the human aorta did not seem to be in the same scale.

We again took the pictures of the normal and AAA lesions in the human aorta, and Fig. 1a was renewed in the revised manuscript.

3. Were there any differences in the length of aorta between control and RhoA cKO mice as shown in Figure 2B?

The length of aorta from the aortic valve to the common iliac artery bifurcation was measured in control and RhoA cKO mice, and the length was not different between the groups. The results were shown in Fig. 2c and described on page 7, lines 18–19 in the revised manuscript.

4. How to explain the phenomenon that AngII treatment increased the expression of α -SMA and F-actin (as shown in Figure 3E-H) but induced AAA both in control and RhoA cKO mice (as shown in Table 1)?

We analyzed control mice in which AAA was formed, and found that the RhoA expression was remarkably reduced in the AAA area compared with the normal area (Supplementary Fig. 3b). This suggests the importance of RhoA for prevention of AAA. Moreover, the reduced expression of α -SMA and impairment of F-actin were also found in the AAA area of the control aorta (Supplementary Fig. 3b). Thus, even in control mice, the expression of RhoA and α -SMA was reduced, and F-actin organization was impaired at the site of AAA. The results were described on page 7, lines 14–17 and page 8, line 25 through page 9, line 1 in the revised manuscript.

5. There are a little grammatical mistakes in the manuscript, please check and correct them.

We thoroughly checked the manuscript and corrected some grammatical and typographical mistakes.

Reviewer #2

- Fig 1. The studies in figure 1 nicely demonstrate that a reduction in RhoA levels is strongly associated with the development of AAA in patients and this correlates with upregulated TIMP/MMP levels (Fig S2). Is there also an association with phosphorylated MAP4K4? It would also be more compelling if some smooth muscle, fibroblast and EC marker levels were shown in the immunoblot (panel H) to get a better feel for the cell types in each extract.

The phosphorylation level of MAP4K4 in the human aorta samples has already been examined and presented in the original manuscript (Fig. 5k–n). As expected, phosphorylated MAP4K4 was up-regulated in the samples from the AAA lesion, compared with those from the normal area. We also confirmed that phosphorylated MAP4K4 was up-regulated and RhoA expression was down-regulated in some control mice in which AAA was formed. These results were shown in Supplementary Fig. 3, and described on page 7, lines 15–18 and page 11, line 25 through page 12, line 1.

We additionally performed the immunoblot for α -SMA, vimentin and CD31, which are smooth muscle, fibroblast and EC marker proteins, respectively. The expression levels of α -SMA and CD31 were reduced in the AAA area due to the disruption of both medial and endothelial layers of the aortic wall, while the level of vimentin was similar between the normal and AAA areas. These results were shown in Fig. 1h–i, and described on page 5, lines 12–16 in the revised manuscript.

- Fig 2 and Fig S1. It is somewhat surprising that depletion of RhoA in SMC compartments from birth has little impact on basal vessel growth and function. While the data provided generally support this conclusion, the statement that blood pressure is not impacted is overstated given that this was performed as a spot check using a tail cuff method. This conclusion should be softened since this method is not sufficient to assess dynamic changes in BP, particularly during the active period- which would require continuous monitoring by radio-telemetry. Also, since the SM22-Cre model also leads to gene inactivation in the developing heart, this should be taken into consideration (see below).

As pointed out by this reviewer, blood pressure measurement was performed as a spot check using a tail-cuff method. This mode of the method was added on page 6, line 1 and page 7, line 12 in the revised manuscript.

Based on the reviewer's comments and previous publications in which the target gene inactivation was shown in the heart of SM22 α -Cre-mediated cKO mice, we examined RhoA expression level in the heart of RhoA cKO adult mice and embryos. The RhoA expression was actually reduced, but not completely lost, in both developing (Embryonic day 11.5 [E11.5]) and adult hearts of RhoA cKO mice, compared with those of control mice. Despite the reduced expression of RhoA in the heart, the cardiac contractility and dimensions in RhoA cKO adult mice were maintained, suggesting that a degree of reduction of cardiac RhoA expression in RhoA cKO mice does not affect the heart function. The results were shown in Supplementary Fig. 2, and described on page 6, line 21 through page 7, line 5 in the revised manuscript.

Nonetheless, since the basal characteristics shown do not change, it would be of interest to explore the extent to which RhoB and C are upregulated to compensate for losses of RhoA.

Since RhoC expression was hardly detected in the mouse aorta, we focused on RhoB expression and found that its expression was increased in the medial layer of the RhoA cKO aorta. As the reviewer expected, this suggests that RhoB might compensate for the loss of RhoA to keep the aortic structure normal in the basal condition. The results were

shown in Supplementary Fig. 1i, and described on page 6, lines 10–13 in the revised manuscript.

Moreover, given the choice of AAA induction that includes application of the lysyl oxidase inhibitor, studies should be done to determine if there are baseline changes in collagen production and collagen cross-linking in the RhoA knockout vessels.

We examined collagen production and collagen cross-linking in the mouse aorta samples by qPCR and hydroxyproline assay as described previously (Ref. 3 in Supplementary Information). The mRNA level of collagen type I $\alpha 1$, the major component of type I collagen, was similar between control and RhoA cKO mice aorta. The hydroxyproline assay also showed that total collagen and cross-linked collagen were equivalently contained in the aorta of both types of mice. The results were shown in Supplementary Fig. 1j–k, and described on page 6, lines 13–19 in the revised manuscript.

- Figure 4. The conclusion that SMC-restricted depletion of RhoA leads to reduced endothelial barrier is overstated. Further studies using dye accumulation to assess barrier function would be required to support this conclusion. Was disruption observed all along the AA or simply at the locale of the aneurism?

Following this reviewer's suggestion, the impaired endothelial barrier function in the RhoA cKO aorta was further confirmed by dye accumulation in the aortic wall. When Evans Blue dye was intravenously injected in the mice, the dye accumulation was significantly increased in AngII+BAPN-treated RhoA cKO mice. The results were shown in Fig. 4b–c, and described on page 9, lines 17–20 in the revised manuscript.

As shown in Fig. 4a, the endothelial layer was disrupted mainly at the site of AAA in RhoA cKO mice after AngII+BAPN treatment. This was also described on page 9, line 16.

- Figure 5. The staining for phosphorylated protein levels in vessel sections is not particularly convincing and would require some control validation studies showing that the signal detected was specific (i.e. reduced following inhibitor treatment or in validated knock-outs).

The specificity of antibodies for immunostaining of phosphorylated ERK1/2, p38 and MAP4K4 was validated by experiments using the RhoA cKO aortas after inhibitor administration. The signals for phosphorylated ERK1/2, p38 and MAP4K4, which were clearly detected by AngII+BAPN treatment, disappeared in the presence of the inhibitor of

each MAP kinase signaling molecule (PD-98059 for ERK1/2, SB-203580 for p38, and DMX-5804 for MAP4K4). The results were shown in Supplementary Fig. 6, and described on page 11, lines 19–23 in the revised manuscript.

- Figure 6. The rescue studies with DMX-5084 are impressive, however controls are missing to show the impact of DMX-5084 on WT treated mice. This is critical to support the conclusions drawn. Also with respect to the comment above that SM22-Cre would be expected to reduce RhoA levels in the heart and prior studies showing that DMX-5084 impacts cardiac contractility, further studies are needed to test this putative interaction in AAA development/progression. Finally, were there any differences in plasma volumes between the treated and un-treated mice of the various genotypes?

The name of the MAP4K4 inhibitor was incorrectly indicated in the original manuscript. The correct name is DMX-5804, but not DMX-5084. We have corrected it in the revised manuscript, and apologize for this mistake.

The results of DMX-5804-treated control mice were added on Table 2, Fig. 6a–c, and Supplementary Fig. 7. Treatment with DMX-5804 did not alter the aortic wall morphology in control mice. There was no change in hemodynamics (Supplementary Fig. 7a). As shown in Supplementary Fig. 2e, cardiac contractility and dimensions evaluated by echocardiography were similar between control and RhoA cKO mice, although RhoA expression in the heart of RhoA cKO was reduced, compared with control mice. In addition, we confirmed that DMX-5804 treatment did not affect the cardiac functions (Supplementary Fig. 7b). Finally, the plasma volume in 1 mL of whole blood was almost same between control and RhoA cKO mice (Supplementary Fig. 7c). These were described on page 12, lines 4–11 in the revised manuscript.

- Figure 7. The mechanistic studies to reveal a direct link between RhoA-SET-MAP4K4-MLCp are too preliminary to support the conclusions drawn. While studies from this manuscript confirm that MAP4K4 is downstream of RhoA – the idea that SET is a major link between active RhoA and MAP4K4 requires further study. Other groups have convincingly shown that RhoA links to MAP4K4 through the well-known RhoA binding partner RhoGDI (Nature Cell Biology volume 21, pages 1565–1577 (2019). Also, SET has previously been shown to be an effector of Rac1 (Oncotarget. 2017 Sep 15; 8(40): 67966–67979). These findings should be integrated into the current studies. Many additional biochemical studies would be necessary to validate SET as a RhoA effector.

To strengthen the link between RhoA–Set–MAP4K4–MLCp, we additionally examined

the effect of Set overexpression on control VSMCs. Even in the presence of RhoA in control VSMCs, overexpressed Set interfered with the PP2A–MAP4K4 association by increasing the PP2A–Set binding. Consequently, phosphorylation of MAP4K4 was enhanced, and downstream of MAP4K4, ERK1/2 and p38 was highly phosphorylated. We further observed that the overexpression suppressed the phosphorylation of MLC2 and facilitated the inactivation (phosphorylation) of MYLK, which may result from the elevated activation of MAP4K4 by the abundant Set-mediated dissociation of MAP4K4 from PP2A. These results were shown in Supplementary Fig. 9, and described on page 15, lines 3–7 and 14–20 in the revised manuscript.

As the reviewer mentioned, the link between RhoA and MAP4K4 was well established in the study published in *Nature Cell Biology*, which has already been cited in the original manuscript (Ref. 34). In the previous *Oncotarget* paper, Set was reported to finely regulate the signaling of Rac1, another Rho family GTPase, and this was additionally described on page 14, line 15 and Ref. 37 in the revised manuscript

Reviewers' comments:

Reviewer #1 (Remarks to the Author):

The author has revised the manuscript as suggested, I have not any additional comments.

Reviewer #2 (Remarks to the Author):

The authors have provided rigorous new data to address nearly all of this reviewer's concerns and should be commended for their efforts. I have two remaining concerns:

1. With respect to the Western blot data provided to address the concern regarding quantitative data showing changes in pMAP4K4 in human samples (Fig 5 m, n) - the figure legend does not clearly state that these data are from human samples. Also, if these are indeed human samples as requested- is this a subset of the samples used in Fig 1H and if so why were these particular samples chosen?

2. The new data provided in response to the previous query regarding RhoB levels is not robust (staining only from one paired sample, no accompanying qPCR or Western analyses or quantification). Given the amount of new data provided to support the other main conclusions of the manuscript, my advice would be to remove these new data (Supplemental Fig 1i) and simply add a line in the discussion indicating the possibility of such compensation.

Response to the Comments from Reviewer #2:

Before responding to this reviewer's comments, we appreciate his/her evaluation for our manuscript. Below is our detailed responses to his/her comments.

Reviewer #2

1. With respect to the Western blot data provided to address the concern regarding quantitative data showing changes in pMAP4K4 in human samples (Fig 5 m, n) - the figure legend does not clearly state that these data are from human samples. Also, if these are indeed human samples as requested- is this a subset of the samples used in Fig 1H and if so why were these particular samples chosen?

We added the words "in human aorta" in the figure legends of the re-revised manuscript to clearly indicate that the data in Fig. 5m and n were obtained from human aorta samples. We selected four samples sequentially from the beginning of each group for western blotting to preliminarily check the level of P-MAP4K4 and total MAP4K4, and found that the results were quite definitive and significantly different between the groups. Thus, we decided to present these results in the manuscript without additional experiments.

2. The new data provided in response to the previous query regarding RhoB levels is not robust (staining only from one paired sample, no accompanying qPCR or Western analyses or quantification). Given the amount of new data provided to support the other main conclusions of the manuscript, my advice would be to remove these new data (Supplemental Fig 1i) and simply add a line in the discussion indicating the possibility of such compensation.

Following this reviewer's advice, the data in Supplementary Fig. 1i were removed from the re-revised manuscript, and the discussion according to the possibility of RhoB expression as compensation mechanism was added in the Discussion section on page 18, lines 21–24 in the re-revised manuscript.